

# Generalized symmetries in F-theory and the topology of elliptic fibrations

Max Hübner[1], David R. Morrison[2], Sakura Schäfer-Nameki[3] and Yi-Nan Wang[4,5]

**1** Department of Physics and Astronomy, University of Pennsylvania, Philadelphia, PA 19104, USA
**2** Departments of Mathematics and Physics, UCSB, Santa Barbara, CA 93106, USA
**3** Mathematical Institute, University of Oxford, Andrew-Wiles Building, Woodstock Road, Oxford, OX2 6GG, UK
**4** School of Physics, Peking University, Beijing 100871, China
**5** Center for High Energy Physics, Peking University, Beijing 100871, China

## Abstract

We realize higher-form symmetries in F-theory compactifications on non-compact elliptically fibered Calabi-Yau manifolds. Central to this endeavour is the topology of the boundary of the non-compact elliptic fibration, as well as the explicit construction of relative 2-cycles in terms of Lefschetz thimbles. We apply the analysis to a variety of elliptic fibrations, including geometries where the discriminant of the elliptic fibration intersects the boundary. We provide a concrete realization of the 1-form symmetry group by constructing the associated charged line operator from the elliptic fibration. As an application we compute the symmetry topological field theories in the case of elliptic three-folds, which correspond to mixed anomalies in 5d and 6d theories.



# 1 Introduction

Generalized Symmetries [1] in string theory compactifications have come to life in recent years [2–22]. The motivation for studying these is at least two-fold: geometric engineering of Quantum Field Theories (QFTs), and the swampland program, in particular the no global symmetry conjecture (for reviews see [23–27]).

In the former, the main motivation is to study QFTs, in particular strongly-coupled theories, which have no weakly coupled Lagrangian description, using dimensional reduction of string theory on non-compact spaces $X$ (which ensures that gravity is decoupled). Generalized symmetries are encoded in the topology of the boundary $\partial X$ of the compactification space. Precisely speaking, relative homology classes give rise to defect operators charged under higher-form symmetries.

On the other hand, the swampland program aims to identify general constraints that a consistent theory of quantum gravity has to satisfy. One of the conjectures is that there are no global symmetries in quantum gravity. This includes not only 0-form symmetries (meaning

ordinary symmetries), but also higher-form symmetries. String theory provides a concrete framework to put these conjectures to a test, by considering dimensional reductions, where the space $X$ is now compact (and thus gravity is not decoupled). To provide evidence or even proof of such a conjecture within string theory, it is crucial to have a characterization of symmetries within the compactification framework, and to understand the symmetry breaking and gauging mechanisms.

A central geometric engineering tool as well as framework for string compactifications is F-theory [28–31]. On one hand F-theory provides a geometric classification of 6d superconformal field theories (SCFTs) [32,33] and a geometric construction of many minimally supersymmetric QFTs in 4d [34,35]. On the other hand it is one of the best understood frameworks for studying string compactifications within the swampland program [36–42]. Understanding the imprint of higher-form symmetries for F-theory compactifications is an important question for both of these programmes.

In M-theory compactifications, which via the M/F-duality are closely related to the F-theory compactifiactions further reduced on a circle, the higher-form symmetries were first discussed in [3, 4] (and subsequently applied in various contexts in [7–11, 13, 14, 16–21]). The main gist of these papers is the identification of the charged operators under a $p$-form symmetry as arising from wrapped M2- and M5-branes on non-compact cycles, modulo compact cycles. The higher-form symmetry is then determined as the Pontryagin dual group. Many of the applications are to generic Calabi-Yau or $G_2$ spaces, where the higher-form symmetry can be computed from the boundary topology, as the torsion part

$$\mathfrak{h}_{(p)} = \text{Tor}\left(\frac{H_p(X, \partial X)}{H_p(X)}\right) \hookrightarrow H_{p-1}(\partial X). \tag{1}$$

In F-theory compactifications on elliptically fibered Calabi-Yau manifolds, it is useful to also first consider the M-theory compactification, as the elliptic fibers are geometrized (as part of the compactification space), and the singularities in the fiber can be resolved. For discriminant components that do not intersect the boundary of the elliptic Calabi-Yau, the situation is similar to M-theory on canonical singularities (without an elliptic fibration), and were studied in [17]. On the contrary when the discriminant intersects the boundary, i.e. when there are non-compact components in the discriminant locus which have the interpretation of flavor branes, the topology of the boundary becomes more intricate, reflecting possible screening effects due to matter fields. Our main goal here is to determine the 1-form symmetries whose charged objects, the line operators, arise from M2-branes wrapped on non-compact 2-cycles in the M-theory compactification. More precisely we first compute the defect group [3,4,43], which is the sum over all $\mathfrak{h}_{(p)}$ in (1).

We propose two approaches to studying this in the context of M/F-theory compactification on elliptic fibrations: by direct analysis of the topology of the boundary, as well as a construction of the relative cycles, tailored specifically to elliptic fibration. Let us briefly summarize the latter: our setup is an elliptically fibered Calabi-Yau $n$-fold $X$, with non-compact base $B$ (which we usually can model in terms of $\mathbb{C}^{n-1}$ or quotients thereof), and an elliptic fiber, whose degenerations are characterized by the discriminant $\Delta$. We will assume throughout that the fibration has a section, and therefore a description as a Weierstrass model. The discriminant vanishes on a (complex) codimension 1 locus in the base $B$, with singular fibers above the generic codimension 1 locus given by the Kodaira classification.

From this structure, we can determine the topology of the elliptic fiber (i.e. $T^2$), and its degeneration to Kodaira singular fibers (and generalizations thereof in higher codimension), and construct non-compact 2-cycles, which are circle fibrations over lines that start at a discriminant component, and stretch to the boundary of $X$. See figure 1. These so-called Lefschetz thimbles are generators of the relative homology groups $\mathfrak{h}_{(2)}$, where the identifica-

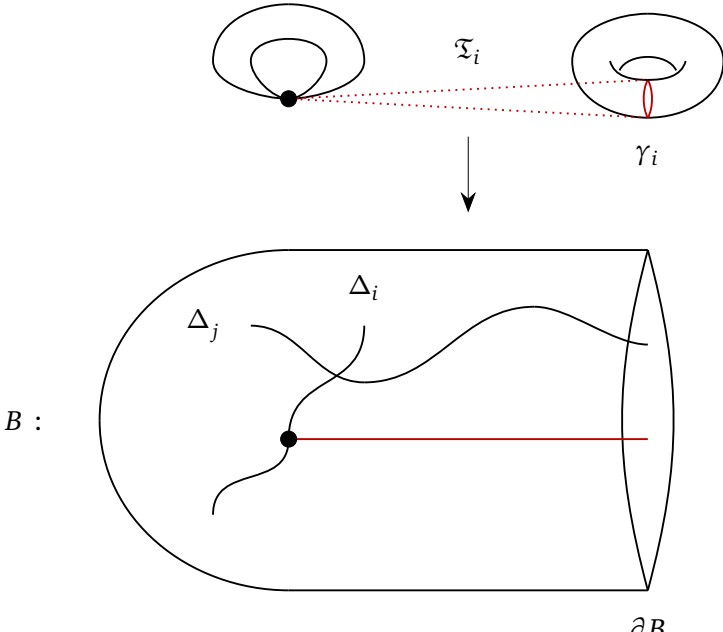

Figure 1: Elliptic Calabi-Yau an $n$-fold $X \to B$ with discriminant $\Delta$ containing a compact and non-compact components denoted $\Delta_i$ and $\Delta_j$ respectively. The Lefschetz thimble $\mathfrak{T}_i \in \mathfrak{h}_{(2)}$ (dotted red) is fibered by a vanishing cycle of $\Delta_i$ and intersects the boundary in $\gamma_i \in H_1(\partial X)$. It projects to a semi-infinite path in the base (red) starting at the discriminant and ending at the boundary of the base.

tions by compact 2-cycles is implemented as an equivalence relation among thimbles. This approach is particularly insightful as it allows systematically to include higher codimension fiber degenerations.

Intuitively, the thimbles (modulo screening) are identified with the generators of the group of line operators. In codimension 1 in the base[1], the singular fibers are of Kodaira type and determine the gauge group in the compactification, and we show that the thimbles generate the (Pontryagin dual of the) center of the gauge group. The thimbles can thus equivalently be represented in terms of *rational* linear combinations of compact curves. The screening is realized as an equivalence with respect to adding integral linear combinations of compact curves (which, when wrapped by M2 branes, realize local operators).

In codimension 2, additional degenerations of the elliptic fiber result in some of the rational curves in the Kodaira fiber becoming reducible. This has the interpretation of matter fields in the dimensional reduction. These additional relations can be systematically characterized using box graphs [44], which in turn provide relations among the thimbles. Codimension three and higher degenerations seem to not change the structure of the thimbles – but of course have implications in the physics of M-theory and F-theory compactifications, e.g. in terms of superpotential couplings.

Our focus is on the 1-form symmetry in M-theory compactifications, which lift to 1-form and "2-form" symmetries[2] in the F-theory uplift. From the intersection of thimbles we also compute contributions to the symmetry TFT [14], which in particular encodes the anomalies for higher-form symmetries. This is applied to various 5d compactifications, that correspond to

---

[1]Here, and throughout this paper, codimension $d$ in in the base means, a complex codimension $d$ sublocus in the base. This e.g. characterizes the sublocus along which $d$ components $\Delta_i$ of the discriminant $\Delta$ of the elliptic fibration vanish simultaneously.

[2]Strictly speaking to identify a 2-form symmetry in 6d, we have to have an absolute theory, i.e. choose a polarization.

duals to 6d SCFTs, such as the non-Higgsable clusters (NHCs) and quivers. In particular these computations point towards the existence of topological couplings in 6d which for absolute theories, become mixed anomalies between 1-form and 2-form symmetries.

The paper is organized as follows. In section 2 we discuss the geometric setup relevant for F-theory, and define the defect group and the realization of the 1-form symmetry in terms of Lefschetz thimbles. In section 3 we then focus on properties of Kodaira thimbles derived from codimension 1 structures of the discriminant. We begin with local K3s and note that every non-compact thimble admits a presentation as a rational collection of compact curves. This further permits us to introduce the divisors Pontryagin dual to Kodaira thimbles, geometrically characterizing 1-form symmetry generators, and we show how these structures persist for general $n$-folds. Here extra screening relations can arise at codimension 2 loci which we study in section 4. These relations derive from the compact curves localized in codimension 2 and we describe how to systematically determine such effects using box graphs. In section 5 we apply the formalism of Kodaira thimbles to the computation of topological couplings describing mixed 't Hooft anomalies among higher-form symmetries. In particular we compute such couplings for all single node NHCs. Finally, in section 6 we study the full defect group for general $n$-folds by analysing the structure of the non-compact cycle in higher degrees, and the topology of the boundary. Among other examples we explicitly consider 6d conformal matter theories, show that although there is no 1-form symmetry for such theories, they can have 3-form symmetries, consistent with the non-simply connected flavor symmetry groups. We end with conclusions and discussions in section 7.

## 2 Defect Groups for Elliptic Fibrations

The 1-form symmetry of a QFT is crucially dependent on the charge lattice of local operators which characterize the possible screening of line operators [1]. In geometric engineering of QFTs from string theory on a space $X$, these local operators in turn are characterized by the geometry $X$. Recent years have seen great progress in determining higher-form symmetries, when $X$ is Calabi-Yau (or $G_2$) [2–22]. In these case the boundary $X$ is smooth and its homology cycles and their intersections determine the higher-form symmetries and more generally the defect group. Typical F-theory geometries fall outside of this class and generically display non-compact singular loci. Geometries with such features are highly interesting as such non-compact loci often signify the presence of flavor symmetries which endow theories with extra structure and can participate in 2-group symmetries. Neither these structures nor 1-form symmetries have been characterized in terms of the boundary topology for such cases. The goal of this paper is to describe 1-form symmetries of 5d/6d theories engineered by elliptically fibered Calabi-Yau $n$-folds with non-compact discriminant loci. We begin by sharpening the questions we wish to address and introducing some background.

### 2.1 The Defect Group of M/F-theory Compactifications

In this paper we study M/F-theory on smooth elliptically fibered Calabi-Yau $n$-folds $\pi : X \to B$ admitting a section $B \to X$. The geometries considered are crepant resolutions of Weierstrass models $W \to B$. The Weierstrass model takes the standard form

$$y^2 = x^3 + f x + g, \tag{2}$$

with $f$ and $g$ sections of $\mathcal{O}(4L)$ and $\mathcal{O}(6L)$ where $L = -K_B$ is the anti-canonical bundle of the Kähler base $B$. The ramification locus of $\pi : X \to B$ is the discriminant locus $\Delta = 4f^3 + 27g^2$ of the Weierstrass model which is possibly reducible with irreducible components $\Delta_i$. The generic elliptic fiber is denoted $\mathbb{E}$.

In this paper we are primarily interested in the gauge theory limit of F-theory, where gravity is decoupled. The base $B$ is thus taken to be a non-compact complex $(n-1)$-dimensional space (oftentimes simply $\mathbb{C}^{n-1}$). The geometric engineering of gauge theories in even dimensions is very well documented in the literature. What is less well-understood are global issues – such as structure of the global gauge and flavor symmetry *groups*, as well as relatedly the higher-form symmetries. This is data that is intrinsically encoded in the non-compact homology classes of the Calabi-Yau space – cycles, which upon wrapping branes, give rise to defect operators in spacetimes [3, 4, 43].

Such non-compact $k$-dimensional homology classes are characterized by the groups

$$\mathfrak{h}_{(k)} = \text{Tor}\left(\frac{H_k(X, \partial X)}{\iota_k(H_k(X))}\right). \tag{3}$$

All homology groups throughout this paper are with integer coefficients. The quotients (3) are computed using the long exact sequence in relative homology of the pair $(X, \partial X)$ which provides the maps

$$
\begin{aligned}
\iota_* : \quad & H_*(X) \to H_*(X, \partial X), \\
J_* : \quad & H_*(\partial X) \to H_*(X).
\end{aligned}
\tag{4}
$$

By exactness we have the alternate characterization of the non-compact homology classes

$$\mathfrak{h}_{(k)} = \ker J_{k-1}, \tag{5}$$

which describes non-compact bulk cycles (i.e. in $X$) as boundary cycles (in $\partial X$), which trivialize when lifted to the bulk. Conversely, we have a contribution to $\mathfrak{h}_{(k)}$ from every vanishing cycle of the bulk which extends non-trivially to the boundary. It is the latter description which will be central to the present paper.

The groups $\mathfrak{h}_{(k)}$ are closely related to the higher-form symmetries of gauge theories obtained upon compactifying F-theory on $X$. Let us start with the discussion in M-theory on the resolved Calabi-Yau $X$ [3, 4] and a general discussion in [7, 19]. We can wrap either M2-branes or M5-branes on non-compact cycles to construct defect operators. The $p$-form symmetries are encoded in

$$\Gamma_{M2}^{(p)} = \mathfrak{h}_{(3-p)}, \qquad \Gamma_{M5}^{(p)} = \mathfrak{h}_{(6-p)}. \tag{6}$$

Technically, the geometry specifies first of all the defect group (as introduced in 6d in [43]), which is obtained as the sum over both M2 and M5 contributions. Then choosing a polarization, i.e. a maximal subset of mutually local defect operators, determines the higher-form symmetries. Unless stated otherwise, we will assume an electric polarization for which $\mathfrak{h}_{(2)}$ characterizes the 1-form symmetry of theory.

Using the standard M/F-theory duality, these wrapped branes on non-compact (relative) cycles, map to branes and strings in F-theory. The M2 and M5 branes wrap non-compact cycles which have the form of a Lefschetz thimble: a compact circle fibered over a non-compact cycle in the base $B$, with the circle collapsing to zero size somewhere on a sublocus (usually complex codim 1) in the base, which is part of the discriminant locus. Applying M/F-duality to such cycles results in the following map: wrapped M2-branes becomes $(p, q)$-strings stretched along the non-compact cycles in the base, ending on a component of the discriminant. Likewise, M5-branes wrapping one of the fiber directions and a non-compact direction in the base, become $(p, q)$ 5-branes.

A note as to what happens, when the M2 and M5 branes wrap the base entirely or the fiber: The M5-branes wrapping the entire elliptic fiber result in D3-branes (with some varying axio-dilaton) or wrapped versions thereof if the M5 wraps in addition subspaces of the base (see [45–47]). We will not consider these further, but they can also potentially interact with higher-symmetries.

The goal of this paper is to determine the defect group from a purely boundary topology analysis of $X$. We will construct representatives of the relative 2-cycles directly from the elliptic fibration. There essentially two ways to proceed: if the discriminant does not intersect with the boundary of $X$, then the monodromy of the elliptic model can be easily computed. Similar analysis was carried out in such instances in [17]. The monodromy perspective can be generalized to the case when the boundary has non-trivial intersection with the discriminant, though the topology of the boundary becomes exceedingly complicated.

An alternative perspective is to construct representatives of the relative 2-cycles using Lefschetz thimbles: these are circle-fibred, with the fiber collapsing above the discriminant loci and admit a presentation as a rational linear combination of compact curves. The latter introduces the standard intersection theoretic tool box into our analysis and we find this approach generalizes to all higher-dimensional Calabi-Yau manifolds. We will now explain the basic idea behind the construction of these thimbles.

## 2.2 Thimbles of Elliptic Fibrations

Consider a non-trivial 1-cycle $\gamma \in H_1(\partial X)$ both in the kernel of the map $H_1(\partial X) \to H_1(X)$ induced by inclusion $\partial X \hookrightarrow X$ and the map $H_1(\partial X) \to H_1(\partial B)$ induced by projection $\partial X \to \partial B$. By the former there exists a relative 2-cycle $\sigma$ in $H_2(X, \partial X)$ restricting to $\gamma$ on the boundary. By the latter $\gamma$ is a non-trivial fibral 1-cycle. Projection of the relative 2-cycle $\sigma$ to the base therefore gives a non-compact connected graph[3] with no loops. Fibral 1-cycles can only collapse at the discriminant locus and therefore end points of this graph necessarily lie on the discriminant locus. The elliptic fibration restricted to this graph is trivial and each edge of the graph can therefore be labelled by a class in $H_1(\mathbb{E})$. The sum of ingoing classes equals the sum of out going classes at internal vertices. Such a graph can be decomposed into a collection of semi-infinite paths labelled by a single class in $H_1(\mathbb{E})$. These describe cycles in $H_2(X, \partial X)$ whose sum returns the relative 2-cycle $\sigma$. The set of relative 2-cycles projecting to paths therefore generates all relative 2-cycles in $H_2(X, \partial X)$ with one leg in the elliptic fibration. This set of generators is however over-complete and our approach to computing $H_2(X, \partial X)$ and more importantly $\mathfrak{h}_{(2)} = H_2(X, \partial X)/H_2(X)$ revolves around understanding such generators and their redundancy relations.

We describe a relative 2-cycle of the above type in more detail. Let $\Gamma$ denote the semi-infinite base path intersecting the discriminant locus $\Delta = \cup \Delta_i$ at a single point $z_i = \Gamma \cap \Delta_i$. Let $\gamma$ be the 1-cycle obtained by restriction of the relative 2-cycle to the boundary $\partial X$. This 1-cycle fibers the relative 2-cycle over $\Gamma \setminus \{z_i\}$. The relative 2-cycle restricted to the fiber $\pi^{-1}(z_i)$ gives a collection of rational curves $\{C_k\}$ where $\pi : X \to B$ is the projection in the resolved model. We denote the relative 2-cycle fixed by this data as

$$\mathfrak{T}'_\Gamma(\gamma, z_i, \{C_k\}) \in Z_2(X, \partial X). \tag{7}$$

Here, $Z_2$ denotes the set of 2-cycles with boundary on $\partial X$. Whenever $z_i$ is a generic point of $\Delta_i$ the curves $\{C_k\}$ are part of the ruling of $\pi^{-1}(\Delta_i)$. The cycle $\mathfrak{T}'_\Gamma(\gamma, z_i, \{C_k\})$ therefore admits continuous deformations to a cycle ending on any point of $\Delta_i$ and all such cycles are homologous only depending on the Kodaira type of the discriminant component $\Delta_i$. This motivates the distinction between what we will call Kodaira thimbles (described above) and Tate thimbles (to be introduced shortly).

---

[3]For local elliptic K3s in F-theory, associated with a collection of $(p, q)$-seven-branes, the physics of the compact and relative 2-cycles can be recast in the framework of string junctions [48–50]. In this frame work such graphs describe multi-pronged string junctions with asymptotic charge. For $n$-folds related relative 2-cycles were studied in [50–53] in deformed geometries. Note further, that for a given configuration we can simply shrink all loops and reduce the graph to a tree.

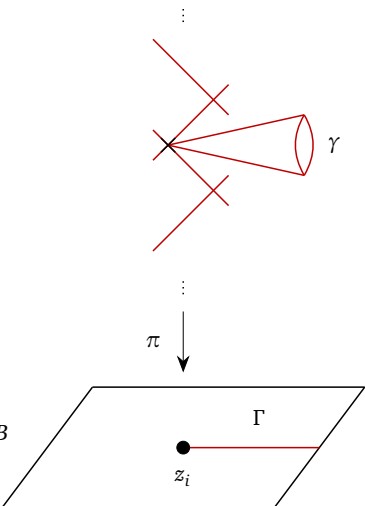

Figure 2: Picture of a representative for the thimble $\mathfrak{T}_i(\gamma) \in \mathfrak{h}_{(2)}$. The thimble projects to the path $\Gamma \subset B$ terminating at $z_i \in \Delta_i$. We depict the fiber $\pi^{-1}(z_i)$ as standard in algebraic geometry with each straight line denoting a rational curve introduced by the resolution.

Let $z_i$ be a generic point of the discriminant component $\Delta_i$. We then call the image of any relative 2-cycle of the form (7) under the projections

$$Z_2(X, \partial X) \to H_2(X, \partial X) \to H_2(X, \partial X)/H_2(X) = \mathfrak{h}_{(2)} \tag{8}$$

a Kodaira thimble and denoted it by

$$\mathfrak{T}_i(\gamma) \in \mathfrak{h}_{(2)}. \tag{9}$$

We sketch a Kodaira thimble in figure 2. Favorably, Kodaira thimbles are independent of the path $\Gamma$ as different choices of paths lead to thimbles differing by compact cycles.

Next, let us introduce Tate thimbles as all generators of $\mathfrak{h}_{(2)}$ resulting from relative cycles (7) under the projection (8) which are not Kodaira thimbles. Tate thimbles capture structures of the elliptic fibration in higher codimension. Kodaira and Tate thimbles do not exhaust $\mathfrak{h}_{(2)}$. We introduce base thimbles as generators for $H_2(B, \partial B)/H_2(B)$ lifted to $X$ via the section $\sigma : B \to X$. These three classes cover all generators of $\mathfrak{h}_{(2)}$ and give the natural splitting

$$\mathfrak{h}_{(2)} = \mathfrak{h}_{f,(2)} \oplus \mathfrak{h}_{b,(2)}, \tag{10}$$

where $\mathfrak{h}_{f,(2)}$ is generated by Kodaira thimbles $\mathfrak{T}_i(\gamma)$ and Tate thimbles capturing data of the elliptic fibration and $\mathfrak{h}_{b,(2)}$ is generated by base thimbles.

Having introduced various thimbles let us discuss redundancies in our description. The map from relative 2-cycles of type (7) to thimbles in $\mathfrak{h}_{f,(2)}$ is many-to-one. Trivially, two such 2-cycles map to the same class in $\mathfrak{h}_{f,(2)}$ whenever they can be continuously deformed into each other. This e.g. occurs when we can slide them along a fixed, connected component of the discriminant. We refer to this as a sliding move, which realizes the homotopy between two such 2-cycles. See figure 3 for a sketch.

The sliding move immediately establishes redundancy relations among Kodaira thimbles associated with different discriminant components. Given two discriminant components $\Delta_i, \Delta_j$ intersecting along $\Delta_{ij} = \Delta_i \cap \Delta_j$ we can slide Kodaira thimbles of both discriminant components onto $\Delta_{ij}$ where they can be compared. We develop this idea further in section 3.1.

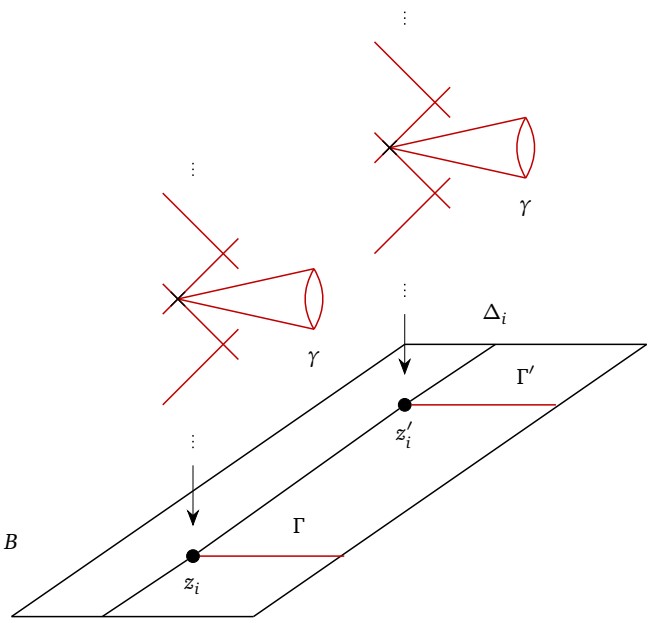

Figure 3: Picture of two homologous relative 2-cycles $\mathfrak{T}'_\Gamma(\gamma, z_i, \{C_k\})$ and $\mathfrak{T}'_{\Gamma'}(\gamma, z'_i, \{C_k\})$. Sliding the thimbles vertically along the discriminant component $\Delta_i$ establishes the homotopy. As a consequence they project to the same thimble $\mathfrak{T}_i(\gamma)$.

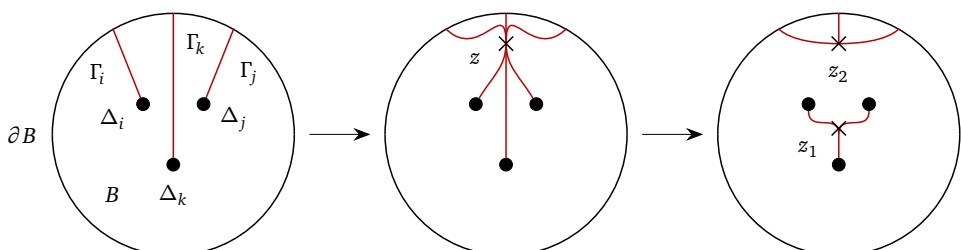

Figure 4: Picture of topological manipulations permitted on (three) thimbles in $\mathfrak{h}_{f,(2)}$ whenever these are linearly dependent. We show the projections of the deformation to the base. The initial configuration (left) are three non-compact thimbles attaching to three disconnected discriminant components. The final configuration (right) lifts to a compact 2-cycle and a non-compact 2-cycle which can be further deformed into $\partial B$.

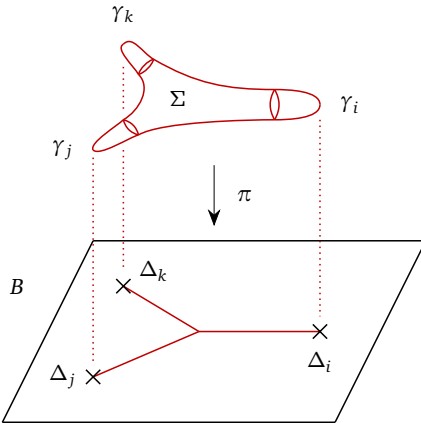

Figure 5: Picture of a compact 2-cycle $\Sigma$ connecting to multiple components of the ramification locus and projecting to a graph.

Another immediate consequence of the sliding move is that we can slide Kodaira thimbles associated with *non-compact* discriminant components to the boundary where they trivialize in relative homology $\mathfrak{h}_{(2)}$. Non-compact discriminant loci therefore imply

$$|\mathfrak{h}_{f,(2)}| \le 1 , \tag{11}$$

as at least one 1-cycle collapses along the restriction of the discriminant to the boundary.

Screening can in addition imply relations between Tate and Kodaira thimbles, which are pairwise topologically distinct in $H_2(X, \partial X)$ and do not necessarily attach to the same connected component of the discriminant locus. Concretely, consider such a collection of thimbles and the associated collection of 1-cycles $\{\gamma_i\}$. Whenever a linear combination of these sum to zero

$$0 = \sum_{i=1}^{n} m^i \gamma_i \in H_1(\partial X) , \tag{12}$$

with integers $m^i \in \mathbb{Z}$, then we have

$$0 = \sum_{i=1}^{n} m^i \mathfrak{T}_i(\gamma_i) \in \mathfrak{h}_{(2)} , \tag{13}$$

as this particular linear combination is homologous to a compact cycle and a non-compact cycle homologous to a cycle contained in the boundary $\partial X$. To show this, deform the associated paths $\Gamma_i$ of such thimbles so that they intersect in a single point $z = \cap_i \Gamma_i$. Now we can separate this junction into two points $z_1 \ne z_2$ by splitting each of the paths $\Gamma_i$ in half. The thimbles split accordingly. The halves of the thimbles associated with path segments of $\Gamma_i$ connecting to $\partial B$ are homologous to cycles contained in $\partial X$ and therefore trivial in $\mathfrak{h}_{(2)}$. The other half of the thimbles connect to discriminant components but are now subsets of a compact 2-cycle which is trivial in $\mathfrak{h}_{(2)}$. We depict this argument in figure 4. We can therefore lift the relation (13) to

$$\sum_{j=1}^{m} \Sigma_j = \sum_{i=1}^{n} m^i \mathfrak{T}_i(\gamma_i) \in H_2(X, \partial X) , \tag{14}$$

with compact 2-cycles $\Sigma_j \in H_2(X)$. We sketch such a compact 2-cycle in figure 5. Each minimal linearly dependent subset of the 1-cycles $\{\gamma_i\}$ contributes such a compact 2-cycle.

# 3 Kodaira Thimbles from Kodaira Fibers

We now explore properties of Kodaira thimbles relevant for general elliptically fibered Calabi-Yau $n$-folds. These derive from codimension 1 structures of the discriminant locus and we therefore begin by considering local K3s. We argue that non-compact 2-cycles, i.e. thimbles, admit presentations as *rational* combinations of compact curves. This reparametrization allows us to introduce similarly rational divisors Pontryagin dual to thimbles. Both are indispensable in the study of anomalies of 1-form symmetries as studied in section 5.

## 3.1 Kodaira Thimbles of local K3s

Let $X$ be local, elliptic K3 surface whose associated Weierstrass model $W \to B$ has a discriminant locus consisting of a single point $z_0 \in B = \mathbb{C}$. The singular fiber $\pi^{-1}(z_0)$ is taken from Kodaira's table of singular fibers and associated with a Lie algebra $\mathfrak{g}$ of rank $r$. Denote by $G$ the simply connected Lie group with Lie algebra $\mathfrak{g}$. The torsional relative 2-cycles of $X$ modulo compact curves are

$$\mathfrak{h}_{(2)} \cong \operatorname{Tor} H_1(\partial X) \cong Z_G \,, \tag{15}$$

where $Z_G$ is the center of $G$. This result is straight forwardly argued from the Mayer-Vietoris long exact sequence, see section 6.1. The generators of $\mathfrak{h}_{(2)}$ are the Kodaira thimbles of $X_2$.

We now argue that Kodaira thimbles admit a presentation as a linear combination of compact curves in $H_2(X)$ with coefficients in $\mathbb{Q}/\mathbb{Z}$. Consider a generator of $H_2(X, \partial X)$ given by the relative 2-cycle $\mathfrak{T}'(\gamma, z_0, \{C_k\})$ which projects to a Kodaira thimble. The intersection matrix of rational curves $C_{\alpha_i}$ generating $H_2(X)$ is the negative of the Cartan matrix of the Lie algebra $\mathfrak{g}$. The intersection vector

$$\mathfrak{T}'(\gamma, z_0, \{C_k\}) \cdot C_{\alpha_i} = w_{\gamma, i} \tag{16}$$

belongs to the weight lattice $\Lambda_{\text{weight}}(\mathbf{R})$ of the representation[4] $\mathbf{R}$ of $\mathfrak{g}$. Varying the collection of rational curves $\{C_k\}$ to which the relative cycle restricts to in $\pi^{-1}(z_0)$ fills the complete lattice. Via the intersection pairing both relative and compact 2-cycles define elements in the space dual to compact 2-cycles $H_2(X)^* = \operatorname{Hom}(H_2(X), \mathbb{Z})$. The latter is a subset of the former. However, the intersection pairing is non-degenerate and therefore any linear form in $H_2(X)^*$ arises as the dual of a linear combination of compact 2-cycles with coefficients in $\mathbb{Q}$. It follows that there exists a rational combination of compact 2-cycles with the same intersection vector (16) as a given relative 2-cycle. We therefore have

$$\mathfrak{T}'(\gamma, z_0, \{C_k\}) = \sum_{i=1}^{r} \beta^i C_{\alpha_i} \,, \qquad \beta^i \in \mathbb{Q} \,, \tag{17}$$

in $H_2(X)^*$ where $\beta^i = \beta^i(\gamma, \{C_k\})$. Varying the collection of curves $\{C_k\}$ shifts the coefficients $\beta^i$ by integers. In fact, the mapping to compact representatives factors through the homology class projection $H_2(X, \partial X) \to H_2(X, \partial X)/H_2(X)$ and we express Kodaira thimbles as

$$\mathfrak{T}_{\mathfrak{g}}(\gamma) = \sum_{i=1}^{r} \beta^i C_{\alpha_i} \,, \qquad \beta^i \in \mathbb{Q}/\mathbb{Z} \,, \tag{18}$$

with coefficients derived from those (17) by evaluating these modulo 1. Kodaira thimbles are labelled by the Lie algebra $\mathfrak{g}$ of the discriminant component they attach to. The compact presentation of thimbles allows us to introduce their self-intersection as the self-intersection of (18) modulo 1.

---

[4]The representations $\mathbf{R}$ is the fundamental representation, the spinor representation, $\mathbf{27}$, $\mathbf{56}$ for $\mathfrak{g} = A_{n-1}, D_{2n+1}, \mathfrak{e}_6, \mathfrak{e}_7$ respectively. For $D_{2n}$ we find two weight vectors $w_{\gamma_j, i}$, with $j = 1, 2$ which belong to the spinor and co-spinor representation.

### 3.1.1 Compact Representatives

We now compute the compact representatives (18) for Kodaira thimbles of local elliptic K3s. Let us denote the negative Cartan matrix of the Lie algebra $\mathfrak{g}$ by $C_{ij} = C_{\alpha_i} \cdot C_{\alpha_j}$. The "Smith normal form" decomposition determines two invertible integer matrices $U, V$ such that

$$C = U \, \text{SNF}(C) \, V \,, \qquad \text{SNF}(C) = \text{diag}\,(1, \ldots, 1, n_1, n_2) \,, \tag{19}$$

where $\mathbb{Z}_{n_1} \times \mathbb{Z}_{n_2} = Z_G$ and $\text{SNF}(C)$ denotes the Smith normal form[5] of the matrix $C$. The columns of $V$ with $n_i \neq 1$ normalized by $n_i$ then determine the rational numbers $\beta^i$ mod 1. We have two thimbles only for $\mathfrak{g} = D_{2n}$. We therefore drop the boundary 1-cycle $\gamma$ from notation when referring to Kodaira thimbles and add superscripts in the case with $\mathfrak{g} = D_{2n}$. We list our labelling conventions of rational curves $C_{\alpha_i}$ together with the compact representatives of Kodaira thimbles and their self-intersections for various simply laced Lie algebras. Similar analyses have appeared in related contexts of 5d SCFTs, i.e. M-theory on (not elliptically fibered) Calabi-Yau three-folds in [9, 14] and in mathematical studies of Lie groups and Lie algebras (cf. [54]). In the present context we will use this method to compute the Kodaira thimbles, which then enter the more intricate analysis for elliptic fibrations in subsequent sections.

**Thimbles of type $A_n$.** With the labelling

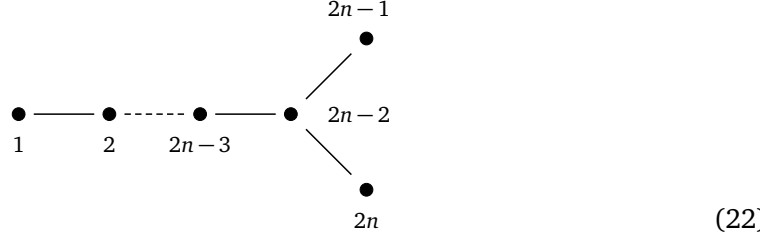

$$\tag{20}$$

we compute

$$\mathfrak{T}_{\mathfrak{su}(n)} = \frac{1}{n} \sum_{i=1}^{n-1} i \, C_{\alpha_i} \,, \tag{21}$$

with self-intersection $\mathfrak{T}_{\mathfrak{su}(n)} \cdot \mathfrak{T}_{\mathfrak{su}(n)} = 1/n$.

**Thimbles of type $D_{2n}$.** With the labelling

$$\tag{22}$$

we compute

$$\mathfrak{T}_{\mathfrak{so}(4n)}^{(c)} = \frac{1}{2} \sum_{i=1}^{n} C_{\alpha_{2i-1}} \,, \qquad \mathfrak{T}_{\mathfrak{so}(4n)}^{(s)} = \frac{1}{2} \sum_{i=1}^{n-1} C_{\alpha_{2i-1}} + \frac{1}{2} C_{\alpha_{2n}} \,. \tag{23}$$

The intersections for $n = 2k$ ($n = 2k+1$) are $\mathfrak{T}_{\mathfrak{so}(4n)}^{(i)} \cdot \mathfrak{T}_{\mathfrak{so}(4n)}^{(j)} = 1/2$ when $i \neq j$ ($i = j$) and zero when $i = j$ ($i \neq j$). Here $c, s$ refer to co-spinor and spinor representations.

---

[5]In principle, a Smith normal form matrix can have more than two nontrivial entries, but this does not happen for Cartan matrices of simply-laced Dynkin diagrams.

**Thimbles of type $D_{2n+1}$.** With the labelling

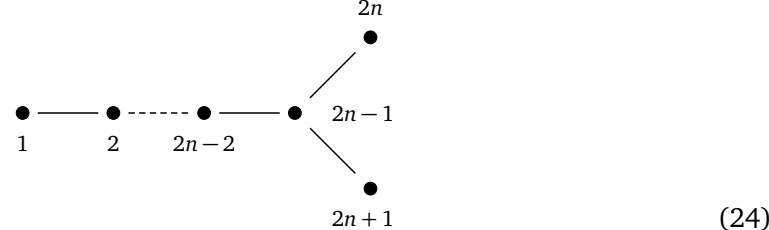

$$\tag{24}$$

we compute

$$\mathfrak{T}_{\mathfrak{so}(4n+2)} = \frac{1}{4}C_{\alpha_{2n+1}} + \frac{3}{4}C_{\alpha_{2n}} + \frac{1}{2}\sum_{i=1}^{n}C_{\alpha_{2i-1}}, \tag{25}$$

with self-intersection $\mathfrak{T}_{\mathfrak{so}(4n+2)} \cdot \mathfrak{T}_{\mathfrak{so}(4n+2)} = 3/4, 1/4$ when $n = 2k, 2k+1$ respectively.

**Thimbles of type $E_6$.** With the labelling

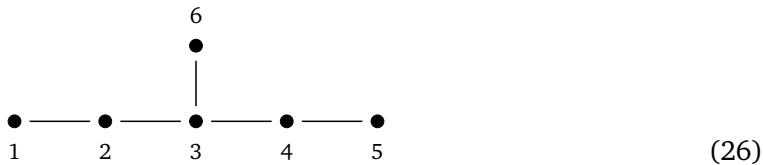

$$\tag{26}$$

we compute

$$\mathfrak{T}_{\mathfrak{e}_6} = \frac{1}{3}(C_{\alpha_1} + 2C_{\alpha_2} + C_{\alpha_4} + 2C_{\alpha_5}), \tag{27}$$

with self-intersection $\mathfrak{T}_{\mathfrak{e}_6} \cdot \mathfrak{T}_{\mathfrak{e}_6} = 2/3$.

**Thimbles of type $E_7$.** With the labelling

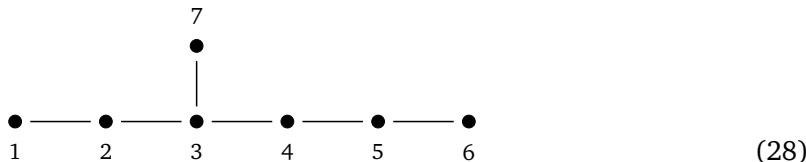

$$\tag{28}$$

we compute

$$\mathfrak{T}_{\mathfrak{e}_7} = \frac{1}{2}(C_{\alpha_4} + C_{\alpha_6} + C_{\alpha_7}), \tag{29}$$

with self-intersection $\mathfrak{T}_{\mathfrak{e}_7} \cdot \mathfrak{T}_{\mathfrak{e}_7} = 1/2$.

**Thimbles of type $E_8$.** For $E_8$ there is no center and thereby no Kodaira thimbles.

The intersections computed above determine a well-known pairing between defects

$$\langle \cdot, \cdot \rangle : \quad \mathfrak{h}_{(2)} \times \mathfrak{h}_{(2)} \rightarrow \mathbb{Q}/\mathbb{Z}, \tag{30}$$

characterizing 't Hooft anomalies between associated higher-form symmetries [13, 15, 55, 56].

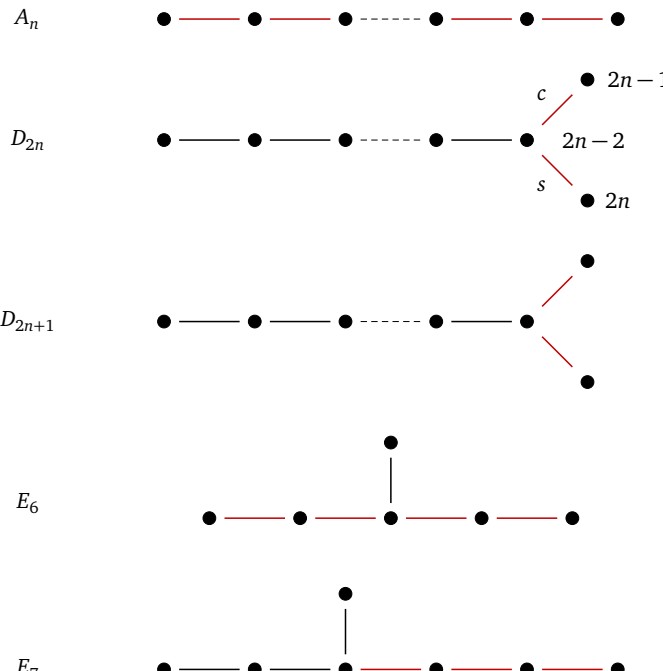

Figure 6: We mark in red the edges in the Dynkin diagrams (identically labelled as those in section 3.1.1) which correspond to points of intersection between to rational curves at which a thimble can end. There are two thimbles for the case $D_{2n}$ associated with the co-spinor, spinor representation. We labelled the edges at which these attach to by $c, s$ respectively.

### 3.1.2 Non-Compact Representatives

The relative 2-cycle $\mathfrak{T}'(\gamma, z_0, \{C_k\}) \in H_2(X, \partial X)$ is the sum of compact 2-cycles $\{C_k\}$ and an irreducible non-compact 2-cycle $\delta$, which is a fibration of a 1-cycle $\gamma$ over a path, where the 1-cycle collapses at one endpoint. The cycle $\delta$ is a Lefschetz thimble and restricting it to $\pi^{-1}(z_0)$ gives a point resulting from the contraction of $\gamma$. The thimbles $\delta$ are the preferred irreducible representatives for Kodaira thimbles and end at the intersection of two rational curves in the resolved Kodaira fiber $C_{\alpha_i}$, and $C_{\alpha_j}$. However not every such intersection point is realized as the end point of a thimble $\delta$. We now characterize thimbles $\delta$ by describing their end points in the resolved fiber.

The intersections of $\mathfrak{T}'(\gamma, z_0, \{C_k\})$ with the curves $C_{\alpha_i}$ produces a weight vectors of a representation $R$ (16). Different weight vectors are realized by distinct collections $\{C_k\}$. The weight system of the representation $R$ can be found for example in [57]. 2-cycles $\delta$ intersect exactly two curves and therefore correspond to weight vectors with exactly two non-vanishing entries, $+1$ and $-1$. Conversely, weight vectors with such entries correspond to a 2-cycle $\delta$ whenever they correspond to edges in the Dynkin diagram. We can therefore determine all possible end points of thimbles $\delta$ in the resolved fiber $\pi^{-1}(z_0)$ by checking which weight vectors match to edges in the Dynkin diagram. We collect our results in figure 6.

### 3.1.3 General Elliptic K3 Surfaces

The compact representatives in section 3.1.1 were computed in the set-up of a local K3 with a single isolated elliptic singularity. For more general set-ups we can ask if the Kodaira thimbles of section 3.1.1 are a sufficient basis for the non-compact 2-cycles of the geometry. We find this not to be the case whenever the discriminant is disconnected. We quantify these effects

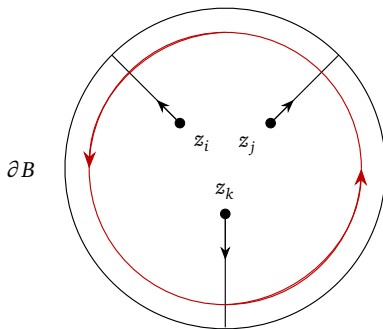

Figure 7: The figure shows three points $z_i, z_j, z_k$ at which the elliptic fiber degenerates. An oriented branch cut emanates from each of these and terminates on the boundary. The topology of $X$ is determined from the total monodromy picked up along the red line.

by computing the thimbles and 1-form symmetries for general elliptic K3s.

Consider an elliptic K3 surface $X_2$ whose associated Weierstrass model $W \to B = \mathbb{C}$ has multiple disconnected discriminant components

$$\Delta = \{z_1, \ldots, z_m\} \,. \tag{31}$$

We compute $\mathfrak{h}_{(2)}$ both using a monodromy and thimble approach. The former is straight forward to apply. The latter again requires us to represent the Kodaira thimbles as elements of $H_2(X)^*$. However $H_2(X)$ now contains compact 2-cycle connecting to multiple discriminant components and projecting to graphs in the base (see figure 5 for an example). These intersect with the compact curves in the fibers $\pi^{-1}(z_i)$ and therefore enter when expanding Kodaira thimbles in terms of compact curves. We begin by computing $\mathfrak{h}_{(2)}$.

**Monodromy.** Let us first consider the monodromy derivation of the defect group $\mathfrak{h}_{(2)}$. For each of the descriminant components we choose non-intersecting branch cuts starting at $z_i$ and ending on $\partial B$, an example is shown in figure 7. Encircling each discriminant component we have a monodromy action $T_{z_i}$. We relabel the discriminant components by the order we encounter their branch cuts along $\partial B$ in counter-clockwise orientation starting from an arbitrary base point on the boundary. The boundary topology is determined from the monodromy

$$T = \prod_{i=1}^{m} T_{z_i} \,. \tag{32}$$

The overall monodromy action for different choices of base point follow from cyclic permutation of the $T_{z_i}$, such permutations are $SL(2, \mathbb{Z})$ equivalent to (32). For this reason $\mathrm{coker}(T-1)$ and therefore also the spectrum of torsional 1-cycles in $\partial X$ are independent of the choice of base point. We have $\mathfrak{h}_{(2)} \cong \mathrm{Tor}\,\mathrm{coker}(T-1)$.

**Thimbles.** Alternatively we can derive $\mathfrak{h}_{(2)}$ via computation of Kodaira thimbles. Given a collection of singular fibers as in (31) we can deform it to a collection of stacks of $(p, q)$-7-branes. The defect group $\mathfrak{h}_{(2)}$ is independent of such deformations. We therefore restrict our attention to two cases functioning as building blocks in more complicated geometries. These involve $m = 2, 3$ mutually local and non-local discriminant components respectively. Further considerations, together with higher-dimensional cases are found in [50–53] where the consequences of deforming the geometry into a collection of such building blocks are studied.

Let us begin by considering the two-component discriminant $\Delta = \{z_1, z_2\}$ with $z_i$ supporting an $I_{N_i}$ singularity. The 1-cycle $\gamma$ collapses at $z_1, z_2$ and traces out a two-sphere $\Sigma$ over a line connecting $z_1, z_2$ with self-intersection $\Sigma \cdot \Sigma = -2$. Let us denote the curves of the fibers $\pi^{-1}(z_i)$ by $C_{\alpha_j}^{N_i}$ and their intersection matrix by $C_{jk}^{N_i} = C_{\alpha_j}^{N_i} \cdot C_{\alpha_k}^{N_i}$ where $j, k = 1, \ldots, N_i - 1$. The intersection matrix between the $N_1 + N_2 - 1$ rational curves of the geometry is

$$
I_{12} = \begin{pmatrix} C_{jk}^{N_1} & 0 & w_1 \\ 0 & C_{jk}^{N_2} & \bar{w}_2 \\ w_1^t & \bar{w}_2^t & -2 \end{pmatrix}, \tag{33}
$$

where $w_1, \bar{w}_2$ are weights of the fundamental and anti-fundamental representation of $\mathfrak{su}(N_i)$ respectively. These weights determined by where $\Sigma$ attaches to in the resolved fiber. For all choices of weights we have

$$
\mathrm{SNF}(I_{12}) = \mathrm{diag}(1, \ldots, 1, N), \tag{34}
$$

where $N = N_1 + N_2$ and therefore

$$
\mathfrak{h}_{(2)} \cong \mathbb{Z}_N. \tag{35}
$$

To compute a Kodaira thimble we now make the Ansatz for a compact representative

$$
\mathfrak{T}_{\mathfrak{su}(N_i)} = \sum_i \beta_i C_i, \qquad \beta_i \in \mathbb{Q}, \tag{36}
$$

for a thimble attaching to the fiber projecting to $z_i$ and now require

$$
\mathfrak{T}_{\mathfrak{su}(N_i)} \cdot C_{\alpha_k}^{N_i} = w_i, \tag{37}
$$

where $w_i$ is again a weight of the fundamental representation of $\mathfrak{su}(N_i)$, with all other intersections vanishing. Generically we find $\mathfrak{T}_{\mathfrak{su}(N_i)}$ to contain the curve $\Sigma$ and not replicate the expansions computed in 3.1.1. In particular note that naively assigning the Kodaira thimbles of section 3.1.1 to the loci $z_i$ can only produce subgroups of $\mathbb{Z}_{N_{\max}}$ where $N_{\max} = \mathrm{lcm}(N_1, N_2)$. This simply follows as both of these thimbles would be order $N_i$ elements, which is clearly incorrect. We conclude that the Kodaira thimbles in section 3.1.1 are not a generating set for the thimbles of K3s with disconnected discriminant loci.

These observations are consistent with the physics of the set-up, in M/F-Theory this geometrically engineers an $\mathfrak{su}(N)$ gauge theory in 7d/8d respectively, which is higgsed as

$$
\mathfrak{su}(N) \rightarrow \mathfrak{su}(N_1) \oplus \mathfrak{su}(N_2) \oplus \mathfrak{u}(1), \tag{38}
$$

and further has massive bifundamental matter with $N$ units of charge under the $\mathfrak{u}(1)$. In M-theory the matter follows from M2 branes wrapped on $\Sigma$ while in F-theory it derives from the open string sector between the two D7 branes. In both cases the matter breaks the center symmetry of the associated simply connected gauge groups from $\mathbb{Z}_{N_1} \times \mathbb{Z}_{N_2} \times U(1)$ to $\mathbb{Z}_N$ matching the geometric result (35).

Next, we consider the three-component discriminant $\Delta = \{z_1, z_2, z_3\}$ with $z_i$ supporting $N_i$ $(p_i, q_i)$-7-branes where $i = 1, 2, 3$. The 1-cycles collapsing at $z_i$ are $\gamma_i = (p_i, q_i) \in \mathbb{Z}^2 \cong H_1(\mathbb{E})$ and are linearly dependent

$$
n_1 \gamma_1 + n_2 \gamma_2 + n_3 \gamma_3 = 0, \qquad n_i = p_k q_j - p_j q_k, \qquad \epsilon_{ijk} = 1. \tag{39}
$$

Whenever $n_i < 0$ we redefine $\gamma_i \rightarrow -\gamma_i$. There exists a single compact 2-cycle $\Sigma$ constructed by fibering the 1-cycles $n_i \gamma_i$ to a common point, see figure 5. The Kodaira thimbles $\mathfrak{T}_{\mathfrak{su}(N_i)}$ are

computed as above from the intersection matrix

$$
\begin{pmatrix}
C^{(1)} & 0 & 0 & w_1 \\
0 & C^{(2)} & 0 & w_2 \\
0 & 0 & C^{(3)} & w_3 \\
w_1^t & w_2^t & w_3^t & \Sigma^2
\end{pmatrix},
\tag{40}
$$

where $w_i$ are some weights of the fundamental representation of $\mathfrak{su}(N_i)$. The self-intersection of the curve $\Sigma$ is computed to

$$
\Sigma \cdot \Sigma = -n_1^2 - n_2^2 - n_3^2 - n_1 n_2 n_3.
\tag{41}
$$

We omit presenting a general formula for the thimbles. Let us discuss the simplest configuration with $N_i = 1$. The only compact 2-cycle of the geometry is $\Sigma$ and in M-theory we find a 7d gauge theory with gauge algebra $\mathfrak{u}(1)$ and a particle of charge $N = \Sigma^2$ obtained by wrapping an M2-brane on $\Sigma$. The unbroken center symmetry is $\mathfrak{h}_{(2)} = \mathbb{Z}_N$ and all thimbles are simply given by $\Sigma/N$.

## 3.2 Kodaira Thimbles for Elliptic Calabi-Yau $n$-folds

We now consider Kodaira thimbles of $n$-folds $X \to B$ with connected discriminant $\Delta = \cup \Delta_i$. We show that the assumption of connectedness is enough to preclude the effects discussed in section 3.1.3 and establish the set of Kodaira thimbles computed for local K3s in section 3.1.1 as a generating set for the Kodaira thimbles of $X$.

We begin with the observation that every compact curve in $H_2(X)$ dualizes via the intersection pairing to an element in

$$
H_{2n-2}(X)^* = \mathrm{Hom}(H_{2n-2}(X), \mathbb{Z}).
\tag{42}
$$

Moreover, we can similarly associate to every non-compact curve in $H_2(X, \partial X)$ an element in $H_{2n-2}(X)^*$. The linear forms constructed from compact cycles $H_2(X)$ are a subgroup of the former. Now consider a non-compact cycle $\mathfrak{T}'_{\mathfrak{g}_i}$ representing a Kodaira thimble $\mathfrak{T}_{\mathfrak{g}_i}$. We can take this cycle to be irreducible following the discussion of section 3.1.2. Clearly this non-compact curve only intersects the Cartan divisors $D^{(i)}_{\alpha_j}$ associated with the discriminant component $\Delta_i$. The presentation of the Kodaira thimble $\mathfrak{T}_{\mathfrak{g}_i}$ as a rational collection of compact curves can therefore only involve compact curves intersecting the divisors $D^{(i)}_{\alpha_j}$. When the discriminant is connected this reduces the possible curves in the expansion of $\mathfrak{T}_{\mathfrak{g}_i}$ to the curves $C^{(i)}_{\alpha_j}$ ruling the divisors $D^{(i)}_{\alpha_j}$. That is we have

$$
\mathfrak{T}_{\mathfrak{g}_i} = \sum_j \beta_{ij} C^{(i)}_{\alpha_j}, \qquad \beta_{ij} \in \mathbb{Q}/\mathbb{Z}.
\tag{43}
$$

This in turn reduces the problem to codimension 1 and we find precisely the coefficients $\beta_{ij}$ of section (3.1.1) where the index $i$ labels for the discriminant component $\Delta_i$.

## 3.3 Pontryagin Dual of Thimbles and Center Divisors

We now introduce the notion of center divisors and divisors Pontryagin dual to Kodaira thimbles of $n$-folds $X \to B$. The latter are not center divisors but crucially fail the integrality condition outlined below only in codimension 2. For the three-fold examples we consider later the center divisors are found to generate the 1-form symmetry group.

Center divisors are rational linear combination of divisors $\mathfrak{D}'$ which have integral intersection numbers with all compact curves of $X$,

$$\text{Center Divisor } \mathfrak{D}' \in H_{2n-2}(X, \mathbb{Q}) \iff \mathfrak{D}' \cdot C \in \mathbb{Z} \quad \forall \, C \in H_2(X). \tag{44}$$

The intersection numbers of center divisors with relative 2-cycles $\mathfrak{T}'_{\mathfrak{g}_i}$ representing Kodaira thimbles are however only rational. The intersection between center divisors and Kodaira thimbles $\mathfrak{T}_{\mathfrak{g}_i}$ are therefore well-defined and non-trivial when taken modulo 1. Intersection numbers between divisors $H_{2n-2}(X)$ and compact curves are integral to begin with and we therefore obtain a well-defined, non-trivial pairing mod 1 between Kodaira thimbles and center divisors in $H_{2n-2}(X, \mathbb{Q}/\mathbb{Z})$

$$\langle \cdot, \cdot \rangle : \quad H_{2n-2}(X, \mathbb{Q}/\mathbb{Z}) \times H_2(X, \mathbb{Q}/\mathbb{Z}) \rightarrow \mathbb{Q}/\mathbb{Z}. \tag{45}$$

This pairing generalizes the one defined in (30) for local K3s to $n$-folds, it determines an action of a center divisor on a Kodaira thimbles with the charge of the action given by their intersection number.

Now we introduce the Pontryagin dual of a Kodaira thimble $\mathfrak{T}_{\mathfrak{g}_i}$ as classes in $H_{2n-2}(X, \mathbb{Q}/\mathbb{Z})$ which only intersects $\mathfrak{T}_{\mathfrak{g}_i}$ among all Kodaira thimbles. Concretely, given the Kodaira thimble

$$\mathfrak{T}_{\mathfrak{g}_i} = \sum_j \beta_{ij} C^{(i)}_{\alpha_j}, \tag{46}$$

we define the Pontryagin dual divisor by replacing the rational curve $C^{(i)}_{\alpha_j}$ with the Cartan divisor $D^{(i)}_{\alpha_j}$ it rules

$$\widehat{\mathfrak{T}}_{\mathfrak{g}_i} = \sum_j \beta_{ij} D^{(i)}_{\alpha_j}, \qquad \beta_{ij} \in \mathbb{Q}/\mathbb{Z}. \tag{47}$$

The pairing between Kodaira thimbles and their Pontryagin dual divisors clearly evaluates to the self-intersections computed for Kodaira thimbles of local K3s in section 3.1.1.

The rational divisors (47) are distinguished by the property that they have vanishing intersection mod 1 with the rational curves ruling any Cartan divisors. They are however not center divisors as they can intersect matter curves in codimension-2 non-integrally. However in all examples we will consider there exist integral combination of divisors of the type (47) which are Pontryagin dual to the generators of $\mathfrak{h}_{(2)}$ and generate the set of center divisors.

## 3.4 Examples: Non-Higgsable-Clusters in 6d

Single node non-Higgsable clusters (NHCs) are Weierstrass models $W_3 \rightarrow B$ with base $B = \mathcal{O}_{\mathbb{P}^1}(-n)$ and $n = 3, 4, 5, 6, 7, 8, 12$ [58]. The base boundary is

$$\partial B = S^3/\mathbb{Z}_n. \tag{48}$$

The coefficient functions $f, g$ and the discriminant $\Delta$ are sections of $-mK_B$ with $m = 4, 6, 12$ respectively and their order of vanishing along the rational curve $C \cong \mathbb{P}^1 \subset \mathcal{O}_{\mathbb{P}^1}(-n)$ follows from the multiplicity $k$ it occurs with in the divisor $-mK_B$. This motivates the ansatz [58]

$$-mK_B = kC + F, \tag{49}$$

with integer $k$ and effective divisor $F$ intersecting non-negatively $C \cdot F \geq 0$. When non-zero the divisor $F$ is non-compact, its a multiple of the fiber class in $\mathcal{O}_{\mathbb{P}^1}(-n)$. The order of vanishing of $f, g, \Delta$ along $C$ was computed in [58] to be

$$[f, g, \Delta] = \left\lceil \frac{m(n-2)}{n} \right\rceil. \tag{50}$$

Consider the expression for $[\Delta]$, here the argument of the ceiling function is integer when $n = 3, 4, 6, 8, 12$ and fractional for $n = 5, 7$. By (49) the latter cases therefore necessarily have non-compact discriminant locus. For the former cases the discriminant is compactly supported on the curve $C$.

Consider the NHCs with $n = 3, 4, 6, 8, 12$. In principle ramification points on the discriminant locus can further ramify the monodromy cover giving rise to non-simply laced gauge algebras. However, at these points the discriminant locus degenerates further [58] contradicting (50). The gauge algebra $\mathfrak{g}$ supported on $C$ of this class of NHCs is therefore simply laced and derives from (50) following Kodaira:

$$
\begin{array}{c||c|c|c|c|c}
n & 3 & 4 & 6 & 8 & 12 \\
\hline
\mathfrak{g} & \mathfrak{su}(3) & \mathfrak{so}(8) & \mathfrak{e}_6 & \mathfrak{e}_7 & \mathfrak{e}_8
\end{array}
\tag{51}
$$

Restriction of the elliptic fibration to a fiber of the line bundle $B = \mathcal{O}_{\mathbb{P}^1}(-n)$ we find the local K3 topology discussed in section 3.1. Kodaira Thimbles attaching to different points of $\mathbb{P}^1$ are homologous and clearly generate $\mathfrak{h}_{f,(2)}$. The discriminant is simply connected with no distinguished points and therefore there exist no further monodromies introducing redundancies among thimbles. As for the local K3 case they are therefore characerized by the centers of the simply connected Lie group with gauge algebra $\mathfrak{g}$. The base thimbles generating $\mathfrak{h}_{b,(2)}$ are associated with $H_1(S^3/\mathbb{Z}_n) \cong \mathbb{Z}_n$. Accounting for both fiber and base thimbles we have:

$$
\begin{array}{c||c|c|c|c|c}
n & 3 & 4 & 6 & 8 & 12 \\
\hline
\mathfrak{h}_{(2)} & \mathbb{Z}_3 \oplus \mathbb{Z}_3 & \mathbb{Z}_2 \oplus \mathbb{Z}_2 \oplus \mathbb{Z}_4 & \mathbb{Z}_3 \oplus \mathbb{Z}_6 & \mathbb{Z}_2 \oplus \mathbb{Z}_8 & \mathbb{Z}_{12}
\end{array}
\tag{52}
$$

For even $n$ and $n = 3$ these results are in agreement with orbifold description for NHCs [3,32]. The NHCs with $n = 5, 7$ have multi-component discriminant loci with non-compact components at which codimension 2 effects enter. We therefore defer their discussion to section 5.

## 4 Kodaira Thimbles for Higher-Codimension Fibers

The analysis thus far focused on singular fibers above co-dimension one of the base, which in terms of the M-/F-theory compatification controls the gauge group. The vanishing order of the discriminant can increase in higher codimension. For codimension 2 in the base, this corresponds to matter fields in the gauge theory, and codimension 3 and 4 to Yukawa couplings (for a review of the geometric engineering dictionary in F-theory, see e.g. [31]). In the context of M-theory on elliptic Calabi-Yau four-folds to 3d $\mathcal{N} = 2$ theories, the presence of CS-interactions could have interesting implications for the generalized symmetries [59].

We now turn to studying the topology of the elliptic fibrations with such higher-codimension singular fibers. The most important changes compared to codimension 1 arise in codimension 2. Field-theoretically, matter can screen 1-form symmetries, and we will see the topological imprint of this effect in the following.

In higher codimension, which in M-/F-theory correspond to couplings in the effective theory, we do not expect any changes in the 1-form symmetry, and this is confirmed by the geometry. This follows as singularities in codimension 3 and higher do not contribute additional curves to the Mori cone $\mathcal{K}$. Indeed, approaching a point of triple intersection along a matter curve one finds the fibral curves to split - just in the codimension 2 case. However, curves introduced by this splitting are already present along some other matter curve. Consequently no new compact curves are produced [44] and no screening relation are added. We will thus focus now on the codimension 2 fibers.

## 4.1 Kodaira Thimbles and 1-Form Symmetry

Consider now a codimension 2 locus in the base of the elliptic fibration. This can e.g. be the intersection of two codimension 1 discriminant loci, or a self-intersection of a single component. Above these loci, the singular fiber degenerates further. We start by considering the Kodaira thimbles $\mathfrak{T}_i$ for each of the codimension 1 fibers, and then analyze, how the presence of the codimension 2 locus changes the analysis of $\mathfrak{h}_{(2)}$. In particular, we will see that a specific linear combination of the codimension 1 Kodaira thimbles will generate $\mathfrak{h}_{(2)}$. Equivalently, a specific combination of the Pontryagin dual divisors $\widehat{\mathfrak{T}}_i$ will generate the 1-form symmetry in the presence of the codimension 2 fiber.

Physically, the matter that arises from M2-branes wrapping rational curves in the fibers introduce relations among line operators, that are realized in terms of M2-branes wrapping the Kodaira thimbles (line operators in the various gauge group factors). This provides a concrete string-theoretic realization of the equivalence relation on the set of genuine line operators $\mathcal{L}$, which defines the Pontryagin dual group to the 1-form symmetry group:

$$\widehat{\Gamma}^{(1)} = \mathcal{L}/\sim, \tag{53}$$

where two lines $L_1, L_2 \in \mathcal{L}$ are in the same equivalence class whenever

$$L_1 \sim L_2 \iff \exists \text{ local operator } \mathcal{O}_{1,2} \text{ which is a junction between } L_1 \text{ and } L_2. \tag{54}$$

This equivalence relation in the QFT realizes the screening of the line operators: if $L_2$ is the trivial line, then $L_1$ is screened.

Line operators in the geometric engineering framework are characterized in terms of non-compact 2-cycles wrapped by M2-branes. We therefore have the identification $\mathcal{L} \cong H_2(X, \partial X)$ with the equivalence between two relative cycles $\mathfrak{T}'_1, \mathfrak{T}'_2$ taking the form

$$\mathfrak{T}'_1 \sim \mathfrak{T}'_2 \iff \exists \text{ compact 2-cycles } C_{1,2} \text{ such that } \mathfrak{T}'_1 = C_{1,2} + \mathfrak{T}'_2, \tag{55}$$

where $C_{1,2}$ is an integral linear combination of effective curves in $\mathcal{K}$. We therefore have

$$\begin{aligned} \widehat{\Gamma}^{(1)} &= \left\{ \mathfrak{T}'_i \in H_2(X, \partial X) \,|\, \text{Thimbles for compact } \Delta_i \right\} / \sim \\ &= \left\{ \mathfrak{T}'_i \in H_2(X, \mathbb{Q}) \,|\, \text{Thimbles for compact } \Delta_i \right\} / \sim, \end{aligned} \tag{56}$$

where we have rewritten relative cycles $\mathfrak{T}'_i$ in terms of their compact representatives. Taking the Pontryagin dual formulation to thimbles, we can now write the 1-form symmetry group $\Gamma^{(1)}$ acting on the lines $\widehat{\Gamma}^{(1)}$. The generators of $\Gamma^{(1)}$ are characterized by divisors Pontryagin dual to thimbles, we have

$$\Gamma^{(1)} = \left\{ \mathfrak{D}'_i \in H_{2n-2}(X, \mathbb{Q}) \,|\, \text{Center divisors for compact } \Delta_i \right\} / \sim, \tag{57}$$

with the center divisor introduced in section 3.3. Here $\sim$ is understood to be the Pontryagin dual equivalence relation to the one appearing in (56), i.e.

$$\mathfrak{D}'_i \sim \mathfrak{D}'_j \iff \mathfrak{D}'_i \cdot C - \mathfrak{D}'_j \cdot C = 0 \pmod 1 \quad \forall \text{ effective curves } C \in \mathcal{K}, \tag{58}$$

which is a necessary requirement for the action of $\Gamma^{(1)}$ on $\widehat{\Gamma}^{(1)}$ to be well-defined.

When computing $\widehat{\Gamma}^{(1)}$ we impose the screening relations (56) in two steps. We first make identifications with curves $C_{1,2}$ ruling Cartan divisors which are associated with codimension 1 structures. This groups $H_2(X, \mathbb{Q})$ into classes spanned by Kodaira thimbles $\mathfrak{T}_i$ associated with compact components $\Delta_i$ of the discriminant. Then we impose screening by the remaining compact curves. In other words, after determining the set of thimbles $\mathfrak{T}_i$ (where screening

in codimension 1 has been accounted for and which can be read off from the discriminant coponents $\Delta_i$) we only have to consider screening effects by codimension-2 matter curves. The Pontryagin dual statement hereof is that we can start with the set of divisors $\widehat{\mathfrak{T}}_i$ and then determine linear combinations of these which intersect all codimension-2 matter curve trivially mod 1.

Note that this perspective will be useful also for studying 2-group symmetries from the perspective of Wilson and flavor lines in the spirit of [11, 13, 60–63]. We will discuss this elsewhere.

## 4.2 Thimbles from Resolution of Singular Fibers

The last section has shown that determining the relative Mori cone $\mathcal{K}$ for the elliptic fibration is essential in order to determine the generalized symmetries. In the following we summarize some very well-known results on elliptic fibrations, and put them into the context of the construction of Kodaira thimbles and associated generalized symmetries.

We will use two approaches to compute $\mathcal{K}$: the direct resolution of the elliptic fibration, as well as the box graph approach [44]. From these, we can determine $\mathfrak{T}$ in a number of examples. Favorably, for all examples considered, the set of Kodaira thimbles $\mathfrak{T}_i$ assigned to each discriminant component indeed give a basis in which the generator $\mathfrak{T}$ of $\mathfrak{h}_{(2)}$ can be expanded as

$$\mathfrak{T} = \sum_i n_i \mathfrak{T}_i \,, \tag{59}$$

with integers $n_i$. In principle Tate thimbles, which can only attach to codimension 2 loci, could enter the expansion (59). However we find the spectrum of Tate thimbles to be trivial for all codimension 2 degenerations considered throughout this section.

The screening effects in codimension 2 are studied for a given resolution of the singular geometry. Schematically, we find a minimal linear combination $\mathfrak{T}$ of the Kodaira thimbles, such that all the 2-cycles in the resolved geometry have integral charge under the corresponding divisor $\widehat{\mathfrak{T}} \in H_{2n-2}(X, \mathbb{Q})$. The residual 1-form symmetry $\Gamma^{(1)}$ is generated by $\mathfrak{T}$. This computational procedure is exactly equivalent to (57) and (58).

Now we comment on the flop invariance of this approach. The divisor $\widehat{\mathfrak{T}}$ and the thimble $\mathfrak{T}$ generating $\mathfrak{h}_{(2)}$ are however independent of the chosen resolution, we claim that $\widehat{\mathfrak{T}}$ and $\mathfrak{T}$ are invariant under flops of $(-1)$-curves. This follows by noting that $(-1)$-curves at codimension 2 loci are labelled by weights of representations of the Lie algebras $\mathfrak{g}_i$ supported on $\Delta_i$. Flopping a $(-1)$-curve replaces it with $(-1)$-curve labeled by a different weight. Two such weights now differ by a collection of roots which are identified in geometry with $(-2)$-curves ruling Cartan divisors [64]. The divisors $\widehat{\mathfrak{T}}_i$ were constructed to precisely intersect such $(-2)$-curves integrally and therefore if a linear combination of generators $\widehat{\mathfrak{T}}_i$ intersects all $(-1)$-curves integrally, then the same linear combination also intersects the $(-1)$ curves in a flopped phase integrally. Given (59) and the Pontryagin dual relation for $\widehat{\mathfrak{T}}$ we find these to be independent under flops.

### 4.2.1 $SU(m) \times SU(n)$ Quivers from Resolutions

We begin by considering some explicit resolutions. In our first example we have a discriminant with two compact irreducible discriminant components, while in our second example one is compact while the other is not. More general situation arise as the combination of these two configurations. In both cases we study the interplay between the Kodaira thimbles associated with each irreducible discriminant component at the codimension 2 locus.

We now consider two compact curves $C_1 = \{u = 0\}$ and $C_2 = \{v = 0\}$ supporting an $I_n$ and $I_m$ singularity respectively intersecting transversely at $u = v = 0$. In M-theory this geometry

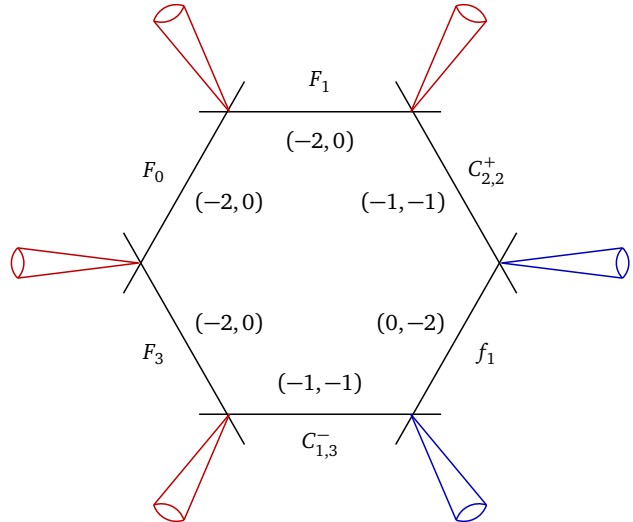

Figure 8: The fiber (black) projecting to the intersection point of an $I_4$ and $I_2$ locus. The pairs $(-n, -m)$ are self intersections within $u = 0$ and $v = 0$ respectively. The normal bundle of curves are either $\mathcal{O}(-2) \oplus \mathcal{O}(0)$ or $\mathcal{O}(-1) \oplus \mathcal{O}(-1)$. We have also depicted the intersection points of Kodaira thimbles of the $I_4$ locus (red) and $I_2$ locus (blue) with the codimension 2 fiber.

engineers a 7d gauge theory with gauge group $SU(m) \times SU(n)$, for electric polarizations, and matter in the bifundamental $(\mathbf{m}, \mathbf{n})$ localized at the intersection point. This matter screens the set of $\mathbb{Z}_m \times \mathbb{Z}_n$ lines defects associated with the gauge algebra factors to a diagonal $\mathbb{Z}_{\gcd(m,n)}$. We now we reproduce this statement from geometry starting with the collision of an $I_4$ and $I_2$ locus. We consider a local geometry and therefore restrict our attention to torsion generators wrapping the fiber $\mathfrak{h}_{f,(2)}$ as defined in (10) and do not specify contributions from the base.

$I_4$ and $I_2$ **Collision.** The local Tate model for a collision of an $I_4$ and $I_2$ locus is

$$y^2 + b_1 x y + b_3 u^2 v y = x^3 + b_2 u x^2 + b_4 u^2 v x + b_6 u^4 v^2. \tag{60}$$

We can choose the resolution sequence

$$(x, y, u; u_1), (x, y, u_1; u_2), (y, u_1; u_3), (x, y, v; v_1), \tag{61}$$

with notation as introduced in [65]. The exceptional divisors of $I_4$ are $u_1, u_2, u_3 = 0$, and the exceptional divisor of $I_2$ is $v_1 = 0$.

The resolved equation is

$$y^2 u_3 + b_1 x y + b_3 u^2 v y u_1 u_3 = x^3 u_1 u_2^2 v_1 + b_2 u x^2 u_1 u_2 + b_4 u^2 v u_1 x + b_6 u^4 v^2 u_1^2 u_3. \tag{62}$$

We denote the fibral curves (rational curves of the Kodaira fiber) by $F_i$, which are one-to-one with the simple roots of the codimension 1 fiber. Starting with the $I_4$ this means $i = 0, \cdots, 3$, where $F_0$ is associated with the affine root. In codimension 2, where the $I_4$ and $I_2$ singularities collide the the fibral $\mathbb{P}^1$ curves are

$$
\begin{aligned}
F_0 &: \quad u = v = 0, \\
F_1 &: \quad u_1 = v = 0, \\
C_{2,2}^+ &: \quad u_2 = v = y u_3 + b_1 x = 0, \\
f_1 &: \quad u_2 = v_1 = 0, \\
C_{1,3}^- &: \quad u_2 = v = y = 0, \\
F_3 &: \quad u_3 = v = 0.
\end{aligned} \tag{63}
$$

This corresponds to the splitting of the fibral curve $F_2$ in codimension 2

$$F_2 \rightarrow C_{2,2}^+ + f_1 + C_{1,3}^-. \tag{64}$$

We depict the codimension 2 fiber in figure 8. From the point of view of the $I_2$ fiber one can likewise obtain the splitting as (denoting by $f_0$ and $f_1$ the two codimension 1 fibers of the $I_2$)

$$f_0 \rightarrow F_0 + F_1 + C_{2,2}^+ + F_3 + C_{1,3}^-. \tag{65}$$

The Kodaira thimbles of the $I_4, I_2$ locus corresponds to the fractional cycles

$$\begin{aligned}
\mathfrak{T}_{\mathfrak{su}(4)} &= \frac{1}{4}(F_1 + 2F_2 + 3F_3), \\
\mathfrak{T}_{\mathfrak{su}(2)} &= \frac{1}{2}f_1,
\end{aligned} \tag{66}$$

respectively. We depict these in figure 8. The Pontryagin dual center divisor are

$$\begin{aligned}
\widehat{\mathfrak{T}}_{\mathfrak{su}(4)} &= \frac{1}{4}(D_1^{(u)} + 2D_2^{(u)} + 3D_3^{(u)}), \\
\widehat{\mathfrak{T}}_{\mathfrak{su}(2)} &= \frac{1}{2}D_1^{(v)},
\end{aligned} \tag{67}$$

with Cartan divisors $D_i^{(u)}, D_1^{(v)}$.

We now consider the screening relations imposed by $C_{2,2}^+$ and $C_{1,3}^-$. For this we compute the charge of the M2 brane wrapping modes over curves under $\widehat{\mathfrak{T}}_{\mathfrak{su}(4)}, \widehat{\mathfrak{T}}_{\mathfrak{su}(2)}$. The M2 brane wrapping mode over the curve $C_{2,2}^+$ has charge

$$C_{2,2}^+ \cdot \widehat{\mathfrak{T}}_{\mathfrak{su}(4)} = -\frac{1}{4}, \qquad C_{2,2}^+ \cdot \widehat{\mathfrak{T}}_{\mathfrak{su}(2)} = \frac{1}{2}. \tag{68}$$

Similar for $C_{1,3}^-$, we have

$$C_{1,3}^- \cdot \widehat{\mathfrak{T}}_{\mathfrak{su}(4)} = \frac{1}{4}, \qquad C_{1,3}^- \cdot \widehat{\mathfrak{T}}_{\mathfrak{su}(2)} = \frac{1}{2}. \tag{69}$$

The 1-form symmetries $\mathbb{Z}_4, \mathbb{Z}_2$ generated individually by $\widehat{\mathfrak{T}}_{\mathfrak{su}(4)}, \widehat{\mathfrak{T}}_{\mathfrak{su}(2)}$ are broken by these fractional charges. Nonetheless, there is a non-trivial linear combination of thimbles and center divisors

$$\mathfrak{T} = 2\mathfrak{T}_{\mathfrak{su}(4)} + \mathfrak{T}_{\mathfrak{su}(2)}, \qquad \widehat{\mathfrak{T}} = 2\widehat{\mathfrak{T}}_{\mathfrak{su}(4)} + \widehat{\mathfrak{T}}_{\mathfrak{su}(2)}, \tag{70}$$

such that all M2 wrapping modes have integral charge under it. Further $2\widehat{\mathfrak{T}} = 2\mathfrak{T} = 0$. Therefore $\mathfrak{h}_{f,(2)} = \mathbb{Z}_2$ generated by $\mathfrak{T}$. Wrapping M2-branes on $\mathfrak{T}$ generates a $\mathbb{Z}_2$ defect group of lines. For a purely electric polarization the divisor $\widehat{\mathfrak{T}}$ therefore generates a $\mathbb{Z}_2$ 1-form symmetry.

One can also take a different resolution that corresponds to a different flop phase (or a different phase of the box graph [44, 66–68]). Nonetheless, the thimble structure and linear combination $\mathfrak{T}$ is unchanged.

The computation presented for the collision of an $I_4$ and $I_2$ fiber straightforwardly generalizes to the case of $I_m, I_n$. Here the thimbles are

$$\mathfrak{T}_{\mathfrak{su}(m)} = \frac{1}{m}\sum_{i=1}^{m-1} iF_i, \tag{71}$$

and

$$\mathfrak{T}_{\mathfrak{su}(n)} = \frac{1}{n}\sum_{i=1}^{n-1} iF_i' \tag{72}$$

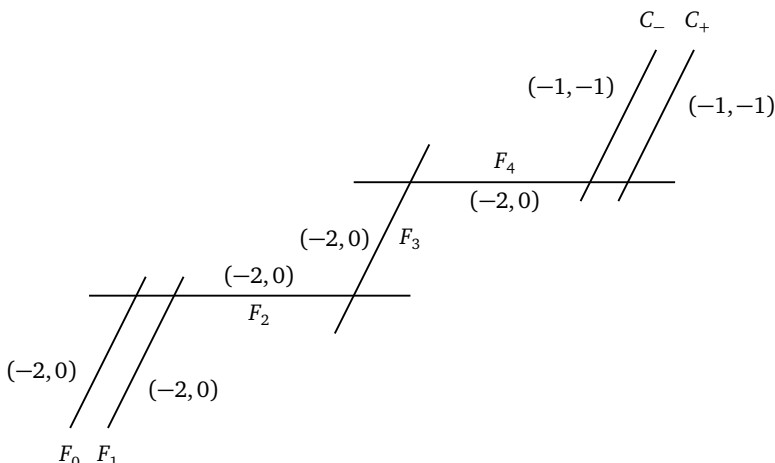

Figure 9: Codimension 2 fiber projecting to the intersection of and $I_1^*$ and $I_1$ fiber.

are combined into

$$\mathfrak{T} = \frac{m}{\gcd(m,n)}\mathfrak{T}_{\mathfrak{su}(m)} + \frac{n}{\gcd(m,n)}\mathfrak{T}_{\mathfrak{su}(n)}, \tag{73}$$

and therefore

$$\mathfrak{h}_{f,(2)} = \mathbb{Z}_{\gcd(m,n)}, \tag{74}$$

which wrapped by M2 branes generates a defect group $\mathbb{Z}_{\gcd(m,n)}$ of lines. For electric polarization we have $\widehat{\mathfrak{T}}$ generating the 1-form symmetry $\mathbb{Z}_{\gcd(m,n)}$. Hence, when $m$ and $n$ are coprime, there is no residual 1-form symmetry.

### 4.2.2   Spin$(4n+2) + 1V$ from Resolutions

Next, we consider the case of an $\mathfrak{so}(4n+2)$ ($n \geq 2$) gauge algebra from type $I_{2n-3}^*$ Kodaira fiber. At a collision point with an $I_1$ locus, the Kodaira fiber type is enhanced to $I_{2n-2}^*$. From the branching rule of $\mathfrak{so}(4n+4) \rightarrow \mathfrak{so}(4n+2) \oplus \mathfrak{u}(1)$,

$$\text{Ad}(\mathfrak{so}(4n+4)) \rightarrow \text{Ad}((\mathfrak{so}(4n+2))_0 + (\mathbf{4n+2})_2 + (\mathbf{4n+2})_{-2} + \mathbf{1}_0, \tag{75}$$

the matter representation at the codimension 2 locus is $1V$. The 1-form symmetry $\mathbb{Z}_4$ from the center of $\mathfrak{so}(4n+2)$ is broken to a subgroup $\mathbb{Z}_2$ by this matter field. We reproduce this result using the resolution geometry.

We consider the case of $n = 2$, that is $\mathfrak{so}(10) + 1V$, and the cases of higher $n$ are completely analogous. The Tate model is

$$y^2 + b_1 uxy + b_3 u^2 vy = x^3 + b_2 ux^2 + b_4 u^3 x + b_6 u^5. \tag{76}$$

In the CY3 case, $b_i$ are complex numbers. The resolution sequence is [65]

$$(x, y, u; u_1), \ (x, y, u_1; u_2), \ (y, u_1; u_3), \ (y, u_2; u_4), \ (u_1, u_3; u_5), \ (u_2, u_3; u_6). \tag{77}$$

The resolved equation is

$$\begin{aligned}
yu_3(yu_4 + b_1 uxu_1 u_2 u_4 u_5 u_6 + b_3 u^2 vu_1 u_5) &= u_1 u_2(x^3 u_2 u_4 u_6 + b_2 ux^2 \\
&\quad + b_4 u^2 u_1 u_3 u_5^2 u_6 x + b_6 u^5 u_1^2 u_3^2 u_5^4 u_6^2).
\end{aligned} \tag{78}$$

The exceptional divisors are $u = 0$, $u_2 = 0$, $u_3 = 0$, $u_4 = 0$, $u_5 = 0$ and $u_6 = 0$. At $v = 0$, the curve $u_4 = 0$ splits into

$$u_4 = v = 0: \quad u_2(b_6 + b_4 x + b_2 x^2) = 0, \tag{79}$$

where we have set the coordinates that cannot vanish in this patch to 1. In the CY case, the above curve splits into three parts $F_2: u_2 = u_4 = 0$ and the two irreducible components $C_+$, $C_-$ of $b_6 + b_4 x + b_2 x^2 = 0$,

$$F_5 = F_4 + C_+ + C_-. \tag{80}$$

In fact $C_+$ and $C_-$ are homologous. The exceptional $\mathbb{P}^1$ curves over the point $v = 0$ are shown in figure 9.

The M2 brane wrapping mode over $C_+$ gives rise to a weight in the vector representation of $\mathfrak{so}(10)$. Now let us consider the thimble of $\mathfrak{so}(10)$:

$$\mathfrak{T}_{\mathfrak{so}(10)} = \frac{1}{4}(2F_1 + 2F_3 + F_4 + 3F_5). \tag{81}$$

The charge of $C_+$ under $\mathfrak{T}_{\mathfrak{so}(10)}$ is

$$C_+ \cdot \mathfrak{T}_{\mathfrak{so}(10)} = -\frac{1}{2}. \tag{82}$$

Hence the $\mathbb{Z}_4$ center symmetry is broken to the subgroup $\mathfrak{h}_{f,(2)} = \mathbb{Z}_2$.

## 4.3 Box Graphs

The discussion of codimension 2 singularities has shown that the essential input determining the defect group of an F-theory compactification is the structure of fibers in codimension 2. If the fiber in codimension 2 corresponds to a local symmetry enhancement $\widetilde{\mathfrak{g}}$, so that the matter is obtained after Higgsing,

$$\widetilde{\mathfrak{g}} \to \mathfrak{g} \oplus \mathfrak{g}_F, \tag{83}$$

where $\mathfrak{g}$ and $\mathfrak{g}_F$ are the gauge and flavor symmetry algebras of the theory, respectively, then this codimension 2 fiber can be described using box graphs introduced in [44, 67, 68] (for an in depth analysis of cases where $\mathfrak{g}_i$ are both non-abelian see [69, 70]).

The box graphs encode the information how the rational curves in the codimension 1 Kodaira singular fibers split at the codimension 2 locus. Denote as before the fibral curves by $F_i$, which are in one-to-one correspondence with the simple roots of $\mathfrak{g}$. Then the box graph contains the information about

$$F_i \to \sum_a F_{i,a}, \tag{84}$$

where $F_{i,a}$ are the codimension 2 fibral curves, as well as the intersections among these (i.e. the codimension 2 fiber).

The box graphs together with the general expression for the thimbles in codimension 1 and two then determines the 1-form symmetry of the F-theory compactification.

### 4.3.1 Example: $SU(n)$-$SU(m)$ Bifundamental Matter

First we consider the $SU(n)$-$SU(m)$ case, i.e. the collison of two codimension 1 discriminant loci, with Kodaira fibers $I_n$ and $I_m$, respectively. This was discussed from the resolution point of view in section 4.2.1, and we will now revisit this from the box graphs. The simplest example we considered was $I_4 - I_2$ collisions. The box graph is a representation graph for the bifundamental matter $(\mathbf{4}, \mathbf{2})$ (e.g. we should think of this as $SU(4)$ gauge group with $N_f = 2$ $\mathbf{4}$ fundamental matter). Denote by

$$(i, j): \qquad \lambda_{i,j} = (L_i^4, L_j^2), \tag{85}$$

the weights and the simple roots are

$$\mathfrak{su}(4): \quad \alpha_i^4 = L_i^4 - L_{i+1}^4, \qquad i = 1, 2, 3$$
$$\mathfrak{su}(2): \quad \alpha^2 = L_1^2 - L_2^2. \tag{86}$$

Then a box graph is a sign assignment $\epsilon_{i,j}$ (usually depicted in terms of the representation graph, and signs $+$ as blue and $-$ as yellow) for each weight which ensures that positive linear combinations of $\epsilon_{i,j}\lambda_{i,j}$ from a cone, and all the simple roots are inside the cone (from a gauge theory point of view, this means these form a consistent Coulomb branch, or geometrically, the associated curves span the relative Mori cone). An example is shown here:

$$
\begin{array}{|c|c|c|c|}
\hline
(1,1) & (1,2) & (1,3) & (1,4) \\
\hline
(2,1) & (2,2) & (2,3) & (2,4) \\
\hline
\end{array}
\tag{87}
$$

The simple roots of $\mathfrak{su}(4)$ act horizontally, the simple root of $\mathfrak{su}(2)$ vertically, as is obvious from the decomposition of the weights (85) A simple rule of thumb for $\mathfrak{su}(n)$ box graphs is that $+$ signs flow up and to the left, $-$ signs down and to the right. The above phase implies that the curve associated to the root $\alpha_2^4$ becomes reducible in codimension 2

$$F_2 \to C_{2,2}^+ + f_1 + C_{1,3}^+, \tag{88}$$

where $C_{i,j}^\epsilon$ is the rational curve, associated to the weight $\lambda_{i,j}$ (i.e. its intersections with Cartan divisors reproduces this weight), and the above box graph corresponds to a fiber where $\epsilon C_{i,j}^\epsilon$ is effective.

In turn we can also determine from the box graph the splitting from the perspective of the codimension 1 $I_2$ fiber[6]

$$f_0 \to F_0 + F_1 + C_{2,2}^+ + C_{1,3}^- + F_3. \tag{89}$$

Note that although this is computed within a given box graph (and thus in the geometry, resolution), the result of this is independent of the box graph. We will prove in examples that these results are independent from the particular Coulomb branch phase (or resolution), i.e. flop-invariant. The box graphs determine generators of the relative Mori cone of the resolve elliptic fibrations, e.g. in this case:

$$\mathcal{K} = \{C_{2,2}^+, f_1, C_{1,3}^-, F_1, F_3\}. \tag{90}$$

Similarly we can determine the splitting of the fiber for $I_m - I_n$ collisions from an $m \times n$ box graph. In this case the codimension 1 thimbles are

$$\mathfrak{T}_{I_m} = \frac{1}{m} \sum_{i=1}^{m-1} i F_i, \tag{91}$$

and

$$\mathfrak{T}_{I_n} = \frac{1}{n} \sum_{i=1}^{n-1} i F_i'. \tag{92}$$

The thimble $\mathfrak{T}$ that is independent (modulo compact curves), in the fiber including the codimension 2 locus is then computed by requiring that

$$\widehat{\mathfrak{T}} \cdot C \in \mathbb{Z}, \qquad \text{for all } C \in \mathcal{K}, \tag{93}$$

---

[6]The splitting of the affine node in an elliptic fiber is a bit more subtle and was discussed in [68].

where $\mathcal{K}$ is the relative Mori cone, as computed from the box graphs. For the $I_n$-$I_m$ example this is the combination

$$\mathfrak{T} = \frac{m}{\gcd(m,n)}\mathfrak{T}_{I_m} + \frac{n}{\gcd(m,n)}\mathfrak{T}_{I_n}. \tag{94}$$

In the above example this is

$$\mathfrak{T} = 2\mathfrak{T}_{\mathfrak{su}(4)} + \mathfrak{T}_{\mathfrak{su}(2)}, \tag{95}$$

which generates a $\mathbb{Z}_2$.

### 4.3.2   Example: $SU(6)+\Lambda^3\mathbf{6}$

The box graphs and associated fiber splittings in codimension 1 for $SU(6)+\Lambda^3\mathbf{6}$ were discussed in [44] – we refer the reader to this paper for the details of the box graph. Locally the model enhances to $\mathfrak{e}_6$, however the fibers are actually monodromy-reduced $\mathfrak{e}_6$ fibers. E.g. one of the box graphs is given by:

$$\tag{96}$$

Each box is a weight $L_{i,j,k} = L_i + L_j + L_k$ $i > j > k$ and the simple roots as before. It was determined that the above corresponds to the splitting

$$F_5 \rightarrow C_{1,2,6}^- + C_{3,4,5}^+ + F_1 + 2F_2 + F_3. \tag{97}$$

The new curves $C^\pm$ are indicated by an $X$ in the above box graph. The relative Mori cone is generated by

$$\mathcal{K} = \{F_1, F_2, F_3, F_4, C_{1,2,6}^-\}. \tag{98}$$

Intersecting the thimble for $\mathfrak{su}(6)$ as e.g. in (91) with all the curves in $\mathcal{K}$. Thus the 1-form symmetry is $\mathbb{Z}_3$.

## 4.4   Codimension 2: Matter-Free Degenerations

In this section we discuss Weierstrass models with torsional sections. These sections generate Mordell-Weil torsion groups whose associated divisors are Pontryagin dual to thimbles. Mordell-Weil torsion in F-theory for global models has been discussed in [71–76]. More specifically we consider the local models for which $SU(N)/\mathbb{Z}_N$ with $N = 2, 3$ is the subgroup of the gauge group $G = SU(N)$ acting faithfully on the representations carried by local operators. In contrast to the examples considered previously we have non-transversely intersecting discriminant components in these cases which are tuned to not introduce extra compact curves in codimension 1 and therefore no additional screening relations arise.

### 4.4.1   $(I_2, I_1)$ with Mordell-Weil Torsion $\mathbb{Z}_2$

Let us first consider the local geometry of intersecting $I_2$ and $I_1$ fibers, with enhancement to type $III$ in codimension 2. The Tate model engineering such a collision is

$$y^2 + a_1 xyz = x^3 + a_2 x^2 z^2 + a_4 xz^4, \tag{99}$$

with associated Weierstrass model

$$y^2 = x^3 + \left(a_4 - \frac{1}{48}(a_1^2 + 4a_2)^2\right)xz^4 - \frac{1}{864}\left((a_1^2 + 4a_2)^2 - 72a_4\right)(a_1^2 + 4a_2)z^6. \tag{100}$$

The discriminant now takes the form

$$\Delta = -\frac{1}{16}\left((a_1^2 + 4a_2)^2 - 64a_4\right)a_4^2, \tag{101}$$

which manifestly has an $I_2$ locus at $a_4 = 0$, and an $I_1$ locus at

$$P := (a_1^2 + 4a_2)^2 - 64a_4 = 0. \tag{102}$$

The resolution of (99) is given by

$$\begin{aligned} (x, y; s),\\ (x, s; t), \end{aligned} \tag{103}$$

and the resolved equation takes the form

$$y^2 s + a_1 x y z s t = x^3 s^2 t^4 + a_2 x^2 z^2 s t^2 + a_4 x z^4. \tag{104}$$

The exceptional divisor for the $SU(2)$ gauge group is $s = 0$. At the intersection point of

$$P = a_4 = 0, \tag{105}$$

the Kodaira $I_2$ fiber is enhanced to type $III$ [71, 75]. There is no new $\mathbb{P}^1$ fiber in this enhancement, hence there is no fundamental matter field under the $\mathfrak{su}(2)$ gauge algebra. There is an additional $\mathbb{Z}_2$ section generating a non-trivial torsional Mordell-Weil subgroup given by $t = 0$.

The Kodaira thimbles for $I_1$ loci are trivial, the only Kodaira thimble of the geometry is therefore $\mathfrak{T}_{\mathfrak{su}(2)}$ associated with the $I_2$ locus. Further, there are no rational curves added to the Mori cone at the codimension 2 point and therefore no additional screening relations to consider, we have

$$\mathfrak{T} = \mathfrak{T}_{\mathfrak{su}(2)}. \tag{106}$$

So whenever the $I_2$ locus is tuned on a compact curve we have

$$\mathfrak{h}_{f,(2)} \cong \mathbb{Z}_2. \tag{107}$$

This is also noted directly from the $III$ fiber which consists of two rational curves meeting at one point of order two. This fiber has no 1-cycle. The 1-cycles which collapse along the $I_1$ and $I_2$ locus must be distinct, the fibers are mutually non-local. It is not possible to deform $\mathfrak{T}_{\mathfrak{su}(2)}$ onto the $I_1$ locus and subsequently slide it off to infinity, i.e. it is indeed a non-trivial non-compact 2-cycle of the geometry.

Next note that the linking pairing on the boundary determines $\operatorname{Tor} H_3(\partial X) \cong \mathbb{Z}_2$. The compact representative of the associated non-compact four-cycle generates this $\mathbb{Z}_2$ torsion group and was determined in [75] to

$$\widehat{\mathfrak{T}} = \widehat{\mathfrak{T}}_{\mathfrak{su}(2)} = \frac{1}{2}S, \tag{108}$$

where $S = \{s = 0\}$ is the resolution divisor of the model (104). This precisely agrees with our definition of $\widehat{\mathfrak{T}}_{\mathfrak{su}(2)}$ given in (47).

Overall we find the geometry to engineer the gauge algebra $\mathfrak{g} = \mathfrak{su}(2)$ and the spectrum of non-compact cycles allows for both the global forms $G = SU(2)$ and $G = SU(2)/\mathbb{Z}_2$ depending on choice of polarization. This conclusion differs from the results of [75], where the gauge group (by requiring it to act faithfully on the representations carried by local operators) was uniquely determined to $G = SU(2)/\mathbb{Z}_2$ for global models. For local models we simply note that the spectrum of local operators is not sufficient to determine the global form of the gauge group, and that non-faithfully acting subgroups of the gauge group depend on a choice of polarization.

### 4.4.2 $(I_3, I_1)$ with Mordell-Weil Torsion $\mathbb{Z}_3$

Now consider the local geometry for an $I_3$ and $I_1$ fiber intersect non-transversely, enhancing to type $IV$ in codimension 2. The Tate model engineering such a collision is

$$y^2 + a_1 xyz + a_3 yz^3 = x^3,\tag{109}$$

with associated Weierstrass model

$$y^2 - x^3 - \frac{1}{48}a_1(a_1^3 - 24a_3)xz^4 + \frac{1}{864}(-a_1^6 + 36a_1^3 a_3 - 216a_3^2)z^6 = 0.\tag{110}$$

The discriminant is

$$\Delta = \frac{1}{16}(27a_3 - a_1^3)a_3^3.\tag{111}$$

The $I_2$ locus is $a_3 = 0$, and there is an $I_1$ locus at

$$Q = a_1^3 - 27a_3 = 0.\tag{112}$$

The resolution of (109) is

$$(x, y; s), \qquad (y, s; p), \qquad (y, p; q).\tag{113}$$

The resolved equation is

$$y^2 s p^2 q^3 + a_1 xyzspq + a_3 yz^3 = x^3 s^2 p,\tag{114}$$

with exceptional divisors for the $SU(3)$ gauge group are $s = 0$ and $p = 0$. At the intersection point of

$$Q = a_3 = 0,\tag{115}$$

the Kodaira $I_3$ fiber is enhanced to type $IV$. There is no new $\mathbb{P}^1$ fiber in this enhancement, hence there is no fundamental matter field under the $SU(3)$ gauge group. As in the previous section we now conclude

$$\mathfrak{h}_{f,(2)} \cong \mathbb{Z}_3.\tag{116}$$

The center divisor Pontryagin dual to the generator $\mathfrak{T}$ of (116) was computed in [75] to

$$\widehat{\mathfrak{T}} = \widehat{\mathfrak{T}}_{\mathfrak{su}(3)} = \frac{1}{3}(S + 2P),\tag{117}$$

where $S = \{s = 0\}$ and $P = \{p = 0\}$. Which is consistent with (116) and again matched by (47).

Overall we find the geometry to engineer the gauge algebra $\mathfrak{g} = \mathfrak{su}(3)$ and the spectrum of non-compact cycles allows for both the global forms $G = SU(3)$ and $G = SU(3)/\mathbb{Z}_3$ depending on the choice of polarization.

## 5 SymTFT for Elliptic Fibrations

There is a multitude of applications of the construction of thimbles. An immediate application is the computation of the so-called symmetry TFT (SymTFT) couplings, discussed in [14]. In M-theory these are – once an absolute theory, i.e. a polarization on the defect group, is chosen – the anomaly theories for (mixed) 't Hooft anomalies. The theories in question are the circle-reductions of 6d SCFTs, and thus KK-theories in 5d. It is these couplings that we will compute here. These couplings are of the type

$$S_{\text{SymTFT}} = c_{ijk} \int B_2^{(i)} B_2^{(j)} B_2^{(k)},\tag{118}$$

where $B_2$ are various background fields for 1-form symmetries, obtained by expanding $G_4$ in suitable 2-forms, dual to compact divisors in the geometry or in our present language, center divisors Pontryagin dual to thimbles (see [14] for a detailed derivation using differential cohomology). There it was shown that the coefficient $c_{ijk}$ depend on the triple intersection numbers of the center divisors in the Calabi-Yau geometry, and the evaluation of the SymTFT in terms of intersection of center divisors generalizes the results relating CS-invariants to intersections of rational linear combinations of curves in [77]. In particular we compute in the following

$$c_{ijk} = \widehat{\mathfrak{T}}^{(i)} \cdot \widehat{\mathfrak{T}}^{(j)} \cdot \widehat{\mathfrak{T}}^{(k)}. \tag{119}$$

These SymTFT topological couplings will uplift via M-/F-duality to topological couplings in the SymTFT for the 6d SCFT. In many instances, these will lift to mixed couplings between background fields of the 1-form and 2-form symmetries (assuming there is an absolute theory). In particular, those $B_2^i$ that are associated with base thimbles lift to 3-form fields $C_3^i$ and fibral thimbles lift to 1-form symmetry backgrounds in 6d. A tensor branch derivation of these mixed anomalies will appear in [78].

## 5.1 Mixed 't Hooft Anomalies for NHCs

Now we compute the center divisors that generate $\mathfrak{h}_{(2)}$, the 1-form symmetry in 5d for the NHCs (51) reduced on a circle. The associated symmetries are uplifted to 1-form and 2-form symmetries in 6d. The resolutions of the singular elliptic CY3 were computed in [79–81]. In the following we use the notation: $\mathbb{F}_n^{b,(N)}$ which is a Hirzebruch $\mathbb{F}_n$ blown up at $b$ points and $(N)$ labels the $N$th compact divisor. We glue these along curves $e, h, f, x_i$ in the notations of [80]. We further label the thimbles by the corresponding center subgroups associated with the discriminant component they attach to.

### 5.1.1 NHCs: $n = 3$

Consider the NHC with $\mathfrak{su}(3)$ on a $(-3)$ curve. The resolution geometry takes the form of

$$
\begin{array}{c}
\mathbb{F}_1^{(3)} \\
e \diagup \quad \diagdown e \\
e \diagup \qquad \diagdown e \\
\mathbb{F}_1^{(1)} \underset{e \quad e}{\rule{2cm}{0.4pt}} \mathbb{F}_1^{(2)}
\end{array}
\tag{120}
$$

The three compact $\mathbb{F}_1$ surfaces $D_1$, $D_2$ and $D_3$ intersect at a common $\mathbb{P}^1$ curve $C_1$ with normal bundle $N_{C_1|X} = \mathcal{O}(-1) \oplus \mathcal{O}(-1)$. Let us denote the fiber curve of $D_1$, $D_2$ and $D_3$ by $C_2$, $C_3$ and $C_4$ respectively. The intersection form $\mathcal{M}_{ij}$ between $D_i$ and $C_j$ are

$$
\begin{array}{c|c|c|c|c}
 & C_1 & C_2 & C_3 & C_4 \\
\hline
D_1 & -1 & -2 & 1 & 1 \\
D_2 & -1 & 1 & -2 & 1 \\
D_3 & -1 & 1 & 1 & -2
\end{array}
\tag{121}
$$

The smith decomposition of the matrix $\mathcal{M}_{ij}$ is

$$
S\mathcal{M}T = \begin{pmatrix} 1 & 0 & 0 & 0 \\ 0 & 3 & 0 & 0 \\ 0 & 0 & 3 & 0 \end{pmatrix}, \quad S = \begin{pmatrix} -1 & -1 & 1 \\ -1 & 0 & 1 \\ 0 & -1 & 1 \end{pmatrix}. \tag{122}
$$

Thus the generators (center divisors) of the two $\mathbb{Z}_3$ subgroups of $\mathfrak{h}_{(2)}$ are

$$\widehat{\mathfrak{T}}_{\mathbb{Z}_3,(1)} = \frac{1}{3}(D_3 - D_1) \, , \, \widehat{\mathfrak{T}}_{\mathbb{Z}_3,(2)} = \frac{1}{3}(D_3 - D_2) \, . \tag{123}$$

One can interpret $D_1$, $D_2$ and $D_3$ as $D_{\alpha_0}$, $D_{\alpha_1}$ and $D_{\alpha_2}$ respectively. Then after the uplifting to 6d, $\widehat{\mathfrak{T}}_{\mathbb{Z}_3,(1)}$ generates the $\mathbb{Z}_3$ 2-form symmetry and $\widehat{\mathfrak{T}}_{\mathbb{Z}_3,(2)}$ generates the $\mathbb{Z}_3$ 1-form symmetry.

We compute the triple intersection numbers among $D_{\mathbb{Z}_3,(i)}$, which gives rise to $B_2^3$ anomaly polynomial in 5d [14] (mod 1).

$$\widehat{\mathfrak{T}}_{\mathbb{Z}_3,(1)}^3 = \widehat{\mathfrak{T}}_{\mathbb{Z}_3,(2)}^3 = 0 \ (\text{mod } 1) \, ,$$
$$\widehat{\mathfrak{T}}_{\mathbb{Z}_3,(1)}^2 \cdot \widehat{\mathfrak{T}}_{\mathbb{Z}_3,(2)} = \widehat{\mathfrak{T}}_{\mathbb{Z}_3,(1)} \cdot \widehat{\mathfrak{T}}_{\mathbb{Z}_3,(2)}^2 = \frac{2}{3} \ (\text{mod } 1) \, . \tag{124}$$

Hence there is a mixed anomaly between the two $\mathbb{Z}_3$ 1-form symmetries in 5d, which translates into a mixed anomaly between the 1-form and 2-form symmetry in 6d.

### 5.1.2 NHCs: $n = 4$

Next consider the NHC $\mathfrak{so}(8)$ on a $(-4)$ curve. The resolution geometry is

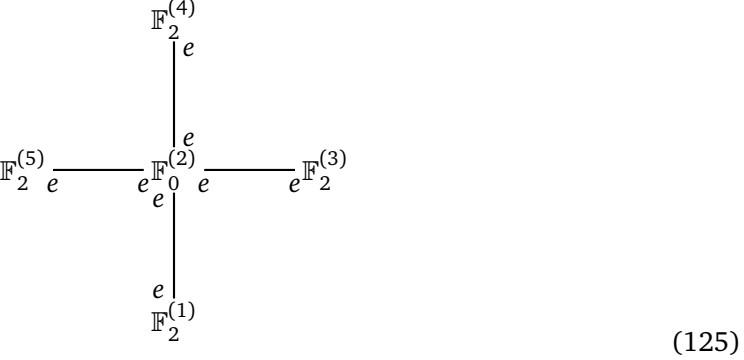

$$\tag{125}$$

Label the compact curves as

$$C_1 = e|_{D_2} \, , \ C_2 = f|_{D_2} \, , \ C_3 = f|_{D_1} \, , \ C_4 = f|_{D_3} \, , \ C_5 = f|_{D_4} \, , \ C_6 = f|_{D_5} \, . \tag{126}$$

The intersection form $\mathcal{M}_{ij} = D_i \cdot C_j$ is

|       | $C_1$ | $C_2$ | $C_3$ | $C_4$ | $C_5$ | $C_6$ |
|-------|-------|-------|-------|-------|-------|-------|
| $D_1$ | 0     | 1     | $-2$  | 0     | 0     | 0     |
| $D_2$ | $-2$  | $-2$  | 1     | 1     | 1     | 1     |
| $D_3$ | 0     | 1     | 0     | $-2$  | 0     | 0     |
| $D_4$ | 0     | 1     | 0     | 0     | $-2$  | 0     |
| $D_5$ | 0     | 1     | 0     | 0     | 0     | $-2$  |

$$\tag{127}$$

As before, we can compute the Smith decomposition, and write down the following generators for the $\mathbb{Z}_2$ and $\mathbb{Z}_4$ factors in $\mathfrak{h}_2 = \mathbb{Z}_2 \oplus \mathbb{Z}_2 \oplus \mathbb{Z}_4$:

$$\widehat{\mathfrak{T}}_{\mathbb{Z}_2,(1)} = \frac{1}{2}(D_5 - D_1) \, , \, \widehat{\mathfrak{T}}_{\mathbb{Z}_2,(2)} = \frac{1}{2}(D_5 - D_3) \, ,$$
$$\widehat{\mathfrak{T}}_{\mathbb{Z}_4} = \frac{1}{4}(3D_1 + 2D_2 + 3D_3 + D_4 - 3D_5) \, . \tag{128}$$

The triple intersection number between them are

$$\widehat{\mathfrak{T}}^3_{\mathbb{Z}_2,(1)} = \widehat{\mathfrak{T}}^3_{\mathbb{Z}_2,(2)} = \widehat{\mathfrak{T}}^3_{\mathbb{Z}_4} = 0 \ (\text{mod } 1)\,,$$

$$\widehat{\mathfrak{T}}^2_{\mathbb{Z}_2,(1)} \cdot \widehat{\mathfrak{T}}_{\mathbb{Z}_2,(2)} = \widehat{\mathfrak{T}}_{\mathbb{Z}_2,(1)} \cdot \widehat{\mathfrak{T}}^2_{\mathbb{Z}_2,(2)} = \widehat{\mathfrak{T}}^2_{\mathbb{Z}_2,(1)} \cdot \widehat{\mathfrak{T}}_{\mathbb{Z}_4} = \widehat{\mathfrak{T}}^2_{\mathbb{Z}_2,(2)} \cdot \widehat{\mathfrak{T}}_{\mathbb{Z}_4} = 0 \ (\text{mod } 1)\,,$$ (129)

$$\widehat{\mathfrak{T}}_{\mathbb{Z}_2,(1)} \cdot \widehat{\mathfrak{T}}^2_{\mathbb{Z}_4} = \widehat{\mathfrak{T}}_{\mathbb{Z}_2,(2)} \cdot \widehat{\mathfrak{T}}^2_{\mathbb{Z}_4} = 0 \ (\text{mod } 1)\,, \quad \widehat{\mathfrak{T}}_{\mathbb{Z}_2,(1)} \cdot \widehat{\mathfrak{T}}_{\mathbb{Z}_2,(2)} \cdot \widehat{\mathfrak{T}}_{\mathbb{Z}_4} = \frac{1}{2} \ (\text{mod } 1)\,.$$

This is a mixed anomaly among the three factors in $\mathfrak{h}_2 = \mathbb{Z}_2 \oplus \mathbb{Z}_2 \oplus \mathbb{Z}_4$.

### 5.1.3 NHCs: $n = 5$

In this section we study the NHC of a single $(-5)$-curve on a two-dimensional base. The tensor branch is

$$\overset{\mathfrak{f}_4}{5}\,,$$ (130)

where there is an $F_4$ gauge group with no matter. The gauge group has a trivial center, and there is no 1-form symmetry in 6d.

The resolution geometry is

$$\mathbb{F}^{(1)}_3 \underset{e}{\rule{2.5em}{0.4pt}}_{h} \mathbb{F}^{(2)}_1 \underset{e}{\rule{2.5em}{0.4pt}}_{e} \mathbb{F}^{(3)}_1 {}_{2h} \underset{}{\rule{2.5em}{0.4pt}}_{e} \mathbb{F}^{(4)}_6 {}_{h} \underset{}{\rule{2.5em}{0.4pt}}_{e} \mathbb{F}^{(5)}_8 \,.$$ (131)

The curves are

$$C_1 = e|_{D_1}\,, \ C_2 = f|_{D_1}\,, \ C_3 = f|_{D_2}\,, \ C_4 = f|_{D_3}\,, \ C_5 = f|_{D_4}\,, \ C_6 = f|_{D_5}\,.$$ (132)

The intersection form $\mathcal{M}_{ij} = D_i \cdot C_j$ is

|       | $C_1$ | $C_2$ | $C_3$ | $C_4$ | $C_5$ | $C_6$ |
|-------|-------|-------|-------|-------|-------|-------|
| $D_1$ | 1     | −2    | 1     | 0     | 0     | 0     |
| $D_2$ | −3    | 1     | −2    | 1     | 0     | 0     |
| $D_3$ | 0     | 0     | 1     | −2    | 1     | 0     |
| $D_4$ | 0     | 0     | 0     | 2     | −2    | 1     |
| $D_5$ | 0     | 0     | 0     | 0     | 1     | −2    |

(133)

After computing the Smith decomposition, the generator for $\mathfrak{h}_2 = \mathbb{Z}_5$ is

$$\widehat{\mathfrak{T}}_{\mathbb{Z}_5} = \frac{1}{5}(-D_1 + 3D_2 + 2D_3 + 3D_4 - D_5)\,.$$ (134)

One can compute

$$\widehat{\mathfrak{T}}^3_{\mathbb{Z}_5} = 0 \ (\text{mod } 1)\,,$$ (135)

hence there is no 't Hooft anomaly for $\mathbb{Z}_5$ itself.

### 5.1.4 NHCs: $n = 6$

The resolution geometry for the $\mathfrak{e}_6$ on $(-6)$ NHC is:

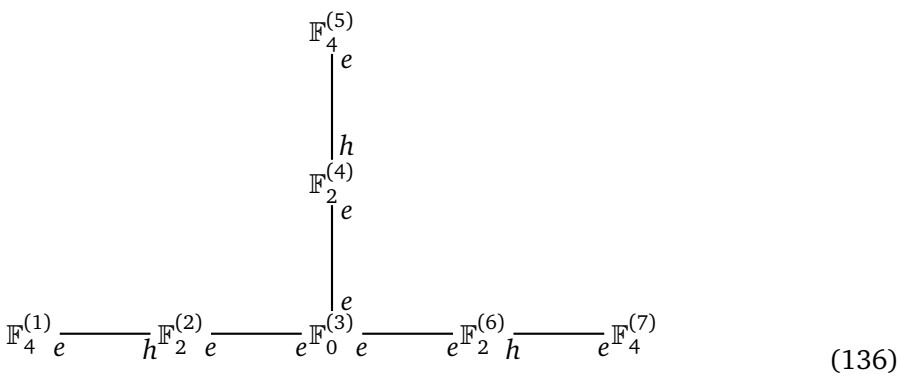

(136)

Label the compact curves as

$$C_1 = e|_{D_3} , \ C_2 = f|_{D_3} , \ C_3 = f|_{D_1} , \ C_4 = f|_{D_2} , \ C_5 = f|_{D_4} , \ C_6 = f|_{D_5} , \ C_7 = f|_{D_6} , \ C_8 = f|_{D_7} . \tag{137}$$

The intersection form $\mathcal{M}_{ij} = D_i \cdot C_j$ is

| | $C_1$ | $C_2$ | $C_3$ | $C_4$ | $C_5$ | $C_6$ | $C_7$ | $C_8$ |
|---|---|---|---|---|---|---|---|---|
| $D_1$ | 0 | 0 | $-2$ | 1 | 0 | 0 | 0 | 0 |
| $D_2$ | 0 | 1 | 1 | $-2$ | 0 | 0 | 0 | 0 |
| $D_3$ | $-2$ | $-2$ | 0 | 1 | 1 | 0 | 1 | 0 |
| $D_4$ | 0 | 1 | 0 | 0 | $-2$ | 1 | 0 | 0 |
| $D_5$ | 0 | 0 | 0 | 0 | 1 | $-2$ | 0 | 0 |
| $D_6$ | 0 | 1 | 0 | 0 | 0 | 0 | $-2$ | 1 |
| $D_7$ | 0 | 0 | 0 | 0 | 0 | 0 | 1 | $-2$ |

$$\tag{138}$$

After computing the Smith decomposition, the generators for the $\mathbb{Z}_3$ and $\mathbb{Z}_6$ factors in $\mathfrak{h}_2 = \mathbb{Z}_3 \oplus \mathbb{Z}_6$ are

$$\widehat{\mathfrak{T}}_{\mathbb{Z}_3} = \frac{1}{3}(D_1 + 2D_2 - 2D_6 - D_7) , \ \widehat{\mathfrak{T}}_{\mathbb{Z}_6} = \frac{1}{6}(3D_1 + 3D_3 - 2D_4 - D_5 + 2D_6 + D_7) . \tag{139}$$

We can compute

$$\widehat{\mathfrak{T}}_{\mathbb{Z}_3}^3 = \widehat{\mathfrak{T}}_{\mathbb{Z}_6}^3 = 0 \ (\text{mod } 1) , \ \widehat{\mathfrak{T}}_{\mathbb{Z}_3}^2 \cdot \widehat{\mathfrak{T}}_{\mathbb{Z}_6} = \widehat{\mathfrak{T}}_{\mathbb{Z}_3} \cdot \widehat{\mathfrak{T}}_{\mathbb{Z}_6}^2 = \frac{2}{3} \ (\text{mod } 1) . \tag{140}$$

Thus there is a mixed anomaly between the $\mathbb{Z}_3$ and $\mathbb{Z}_6$ 1-form symmetry in 5d, which uplifts to a mixed anomaly between the $\mathbb{Z}_3$ 1-form and $\mathbb{Z}_6$ 2-form symmetry in 6d.

### 5.1.5 NHCs: $n = 7$

For the NHC of a single $(-7)$-curve on a two-dimensional base. The tensor branch is

$$\overset{\mathfrak{e}_7}{7} , \tag{141}$$

where there is an $\mathfrak{e}_7$ gauge algebra with a fundamental half-hypermultiplet $\frac{1}{2}\mathbf{56}$. The 1-form symmetry of $E_7$ is completely broken by the presence of fundamental matter field, and there is no 1-form symmetry in 6d.

The resolution geometry is

$$\tag{142}$$

The curves are

$$C_1 = e|_{D_1} , \ C_2 = f|_{D_1} , \ C_3 = f|_{D_2} , \ C_4 = f|_{D_3} , \ C_5 = f|_{D_4} , \ C_6 = f|_{D_5} , \ C_7 = x_1|_{D_5} ,$$
$$C_8 = f|_{D_6} , \ C_9 = f|_{D_7} , \ C_{10} = f|_{D_8} . \tag{143}$$

The intersection form $\mathcal{M}_{ij} = D_i \cdot C_j$ is

|       | $C_1$ | $C_2$ | $C_3$ | $C_4$ | $C_5$ | $C_6$ | $C_7$ | $C_8$ | $C_9$ | $C_{10}$ |
|-------|-------|-------|-------|-------|-------|-------|-------|-------|-------|----------|
| $D_1$ | 3     | −2    | 1     | 0     | 0     | 0     | 0     | 0     | 0     | 0        |
| $D_2$ | −5    | 1     | −2    | 1     | 0     | 0     | 0     | 0     | 0     | 0        |
| $D_3$ | 0     | 0     | 1     | −2    | 1     | 0     | 0     | 0     | 0     | 0        |
| $D_4$ | 0     | 0     | 0     | 1     | −2    | 1     | 0     | 0     | 1     | 0        |
| $D_5$ | 0     | 0     | 0     | 0     | 1     | −2    | −1    | 1     | 0     | 0        |
| $D_6$ | 0     | 0     | 0     | 0     | 0     | 1     | 1     | −2    | 0     | 1        |
| $D_7$ | 0     | 0     | 0     | 0     | 1     | 0     | 1     | 0     | −2    | 0        |
| $D_8$ | 0     | 0     | 0     | 0     | 0     | 0     | −1    | 1     | 0     | −2       |

$$\tag{144}$$

After computing the Smith decomposition, the generator for $\mathfrak{h}_2 = \mathbb{Z}_7$ is

$$\widehat{\mathfrak{T}}_{\mathbb{Z}_7} = \frac{1}{7}(D_1 + 2D_2 + 3D_3 + 4D_4 + 3D_5 + 2D_6 + 2D_7 + D_8). \tag{145}$$

One can compute

$$\widehat{\mathfrak{T}}_{\mathbb{Z}_7}^3 = 0 \,(\mathrm{mod}\,1), \tag{146}$$

hence there is no 't Hooft anomaly for $\mathbb{Z}_7$ itself.

### 5.1.6 NHCs: $n = 8$

The resolution geometry for $\mathfrak{e}_7$ is:

$$
\begin{array}{c}
\mathbb{F}_2^{(5)} \\
\uparrow e \\
\;\; \Big| \\
e \\
\mathbb{F}_6^{(1)} \underset{e}{\quad} \overset{h}{\quad} \mathbb{F}_4^{(2)} \underset{e}{\quad} \overset{h}{\quad} \mathbb{F}_2^{(3)} \underset{e}{\quad} \overset{e}{\quad} \mathbb{F}_0^{(4)} \underset{e}{\quad} \overset{e}{\quad} \mathbb{F}_2^{(6)} \underset{h}{\quad} \overset{e}{\quad} \mathbb{F}_4^{(7)} \underset{h}{\quad} \overset{e}{\quad} \mathbb{F}_6^{(8)}
\end{array}
\tag{147}
$$

Label the compact curves as

$$C_1 = e|_{D_4}\,,\; C_2 = f|_{D_4}\,,\; C_3 = f|_{D_1}\,,\; C_4 = f|_{D_2}\,,\; C_5 = f|_{D_3}\,,\; C_6 = f|_{D_5}\,,\; C_7 = f|_{D_6}\,,$$
$$C_8 = f|_{D_7}\,,\; C_9 = f|_{D_8}\,. \tag{148}$$

The intersection form $\mathcal{M}_{ij} = D_i \cdot C_j$ is

|       | $C_1$ | $C_2$ | $C_3$ | $C_4$ | $C_5$ | $C_6$ | $C_7$ | $C_8$ | $C_9$ |
|-------|-------|-------|-------|-------|-------|-------|-------|-------|-------|
| $D_1$ | 0     | 0     | −2    | 1     | 0     | 0     | 0     | 0     | 0     |
| $D_2$ | 0     | 0     | 1     | −2    | 1     | 0     | 0     | 0     | 0     |
| $D_3$ | 0     | 1     | 0     | 1     | −2    | 0     | 0     | 0     | 0     |
| $D_4$ | −2    | −2    | 0     | 0     | 1     | 1     | 1     | 0     | 0     |
| $D_5$ | 0     | 1     | 0     | 0     | 0     | −2    | 0     | 0     | 0     |
| $D_6$ | 0     | 1     | 0     | 0     | 0     | 0     | −2    | 1     | 0     |
| $D_7$ | 0     | 0     | 0     | 0     | 0     | 0     | 1     | −2    | 1     |
| $D_8$ | 0     | 0     | 0     | 0     | 0     | 0     | 0     | 1     | −2    |

$$\tag{149}$$

After computing the Smith decomposition, the generators for the $\mathbb{Z}_2$ and $\mathbb{Z}_8$ factors in $\mathfrak{h}_2 = \mathbb{Z}_2 \oplus \mathbb{Z}_8$ are

$$\widehat{\mathfrak{T}}_{\mathbb{Z}_2} = \frac{1}{2}(D_1 + D_3 + D_5)\,,\; \widehat{\mathfrak{T}}_{\mathbb{Z}_8} = \frac{1}{8}(3D_1 - 2D_2 + D_3 + 4D_4 - 2D_5 + D_6 - 2D_7 + 3D_8). \tag{150}$$

We can compute

$$\widehat{\mathfrak{T}}_{\mathbb{Z}_2}^3 = \widehat{\mathfrak{T}}_{\mathbb{Z}_8}^3 = 0 \ (\text{mod } 1) \,, \ \widehat{\mathfrak{T}}_{\mathbb{Z}_2}^2 \cdot \widehat{\mathfrak{T}}_{\mathbb{Z}_8} = \frac{1}{2} \ (\text{mod } 1) \,, \ \widehat{\mathfrak{T}}_{\mathbb{Z}_2} \cdot \widehat{\mathfrak{T}}_{\mathbb{Z}_8}^2 = 0 \ (\text{mod } 1) \,. \tag{151}$$

Thus there is a mixed anomaly between the $\mathbb{Z}_2$ and $\mathbb{Z}_8$ 1-form symmetry in 5d, which uplifts to a mixed anomaly between the $\mathbb{Z}_2$ 1-form and $\mathbb{Z}_8$ 2-form symmetry in 6d.

### 5.1.7  NHCs: $n = 12$

The resolution geometry for $\mathfrak{e}_8$ is:

$$\mathbb{F}_{10}^{(1)} \underset{e}{\rule{1.5cm}{0.4pt}} {}_h \mathbb{F}_8^{(2)} \underset{e}{\rule{1.5cm}{0.4pt}} {}_h \mathbb{F}_6^{(3)} \underset{e}{\rule{1.5cm}{0.4pt}} {}_h \mathbb{F}_4^{(4)} \underset{e}{\rule{1.5cm}{0.4pt}} {}_h \mathbb{F}_2^{(5)} \underset{e}{\rule{1.5cm}{0.4pt}} {}_e \mathbb{F}_0^{(6)} \underset{e}{\rule{1.5cm}{0.4pt}} {}_e \mathbb{F}_2^{(8)} {}_h \underset{e}{\rule{1.5cm}{0.4pt}} \mathbb{F}_4^{(9)}$$

with $\mathbb{F}_2^{(7)}$ attached above $\mathbb{F}_0^{(6)}$ via $e$.

$$\tag{152}$$

Label the compact curves as

$$C_1 = e|_{D_6} \,, \ C_2 = f|_{D_6} \,, \ C_3 = f|_{D_1} \,, \ C_4 = f|_{D_2} \,, \ C_5 = f|_{D_3} \,, \ C_6 = f|_{D_4} \,, \ C_7 = f|_{D_5} \,,$$
$$C_8 = f|_{D_7} \,, \ C_9 = f|_{D_8} \,, \ C_{10} = f|_{D_9} \,. \tag{153}$$

The intersection form $\mathcal{M}_{ij} = D_i \cdot C_j$ is

|       | $C_1$ | $C_2$ | $C_3$ | $C_4$ | $C_5$ | $C_6$ | $C_7$ | $C_8$ | $C_9$ | $C_{10}$ |
|-------|-------|-------|-------|-------|-------|-------|-------|-------|-------|----------|
| $D_1$ | 0 | 0 | $-2$ | 1 | 0 | 0 | 0 | 0 | 0 | 0 |
| $D_2$ | 0 | 0 | 1 | $-2$ | 1 | 0 | 0 | 0 | 0 | 0 |
| $D_3$ | 0 | 0 | 0 | 1 | $-2$ | 1 | 0 | 0 | 0 | 0 |
| $D_4$ | 0 | 0 | 0 | 0 | 1 | $-2$ | 1 | 0 | 0 | 0 |
| $D_5$ | 0 | 1 | 0 | 0 | 0 | 1 | $-2$ | 0 | 0 | 0 |
| $D_6$ | $-2$ | $-2$ | 0 | 0 | 0 | 0 | 1 | 1 | 1 | 0 |
| $D_7$ | 0 | 1 | 0 | 0 | 0 | 0 | 0 | $-2$ | 0 | 0 |
| $D_8$ | 0 | 1 | 0 | 0 | 0 | 0 | 0 | 0 | $-2$ | 1 |
| $D_9$ | 0 | 0 | 0 | 0 | 0 | 0 | 0 | 0 | 1 | $-2$ |

$$\tag{154}$$

The center divisor for $\mathfrak{h}_2 = \mathbb{Z}_{12}$ is

$$\widehat{\mathfrak{T}}_{\mathbb{Z}_{12}} = \frac{1}{12}(-D_1 - 2D_2 - 3D_3 - 4D_4 - 5D_5 + 6D_6 - 3D_7 - 4D_8 - 2D_9) \,. \tag{155}$$

We have

$$\widehat{\mathfrak{T}}_{\mathbb{Z}_{12}}^3 = 0 \ (\text{mod } 1) \,. \tag{156}$$

The 1-form symmetry $\mathbb{Z}_{12}$ has no 't Hooft anomaly.

## 5.2  Example: Spin$-$Sp Quiver

There is a large class of quivers in 6d, which have 1-form symmetry and defect group (see [13] for a systematic way to compute these). A nice set of examples are 6d Spin$-Sp$ quivers. We consider the simplest case of an Spin(10) gauge group on a $(-4)$-curve:

$$\overset{\mathfrak{so}(10)}{4} - [\mathfrak{sp}(2)] \,. \tag{157}$$

The resolution geometry is

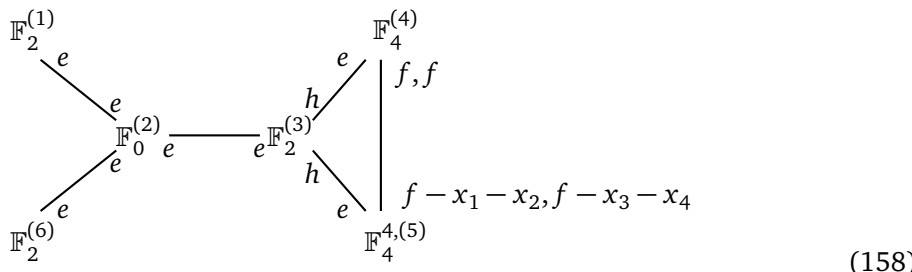

$$\tag{158}$$

Note that $D_5$ is obtained by blowing up $\mathbb{F}_4$ four times, and the exceptional curves are $x_i$ $(i = 1, \ldots, 4)$.

Label the compact curves as

$$C_1 = e|_{D_2} \,,\ C_2 = f|_{D_2} \,,\ C_3 = f|_{D_1} \,,\ C_4 = f|_{D_6} \,,\ C_5 = f|_{D_3} \,,\ C_6 = f|_{D_4} \,,\ C_7 = x_1|_{D_5} \,,$$
$$C_8 = x_2|_{D_5} \,,\ C_9 = x_3|_{D_5} \,,\ C_{10} = x_4|_{D_5} \,. \tag{159}$$

The intersection form $\mathcal{M}_{ij} = D_i \cdot C_j$ is

|  | $C_1$ | $C_2$ | $C_3$ | $C_4$ | $C_5$ | $C_6$ | $C_7$ | $C_8$ | $C_9$ | $C_{10}$ |
|---|---|---|---|---|---|---|---|---|---|---|
| $D_1$ | 0 | 1 | $-2$ | 0 | 0 | 0 | 0 | 0 | 0 | 0 |
| $D_2$ | $-2$ | $-2$ | 1 | 1 | 1 | 0 | 0 | 0 | 0 | 0 |
| $D_3$ | 0 | 1 | 0 | 0 | $-2$ | 1 | 0 | 0 | 0 | 0 |
| $D_4$ | 0 | 0 | 0 | 0 | 1 | $-2$ | 1 | 1 | 1 | 1 |
| $D_5$ | 0 | 0 | 0 | 0 | 1 | 0 | $-1$ | $-1$ | $-1$ | $-1$ |
| $D_6$ | 0 | 1 | 0 | $-2$ | 0 | 0 | 0 | 0 | 0 | 0 |

$$\tag{160}$$

The center divisors for $\mathfrak{h}_2 = \mathbb{Z}_2 \times \mathbb{Z}_4$ are

$$\widehat{\mathfrak{T}}_{\mathbb{Z}_2} = \frac{1}{2}(D_1 + D_6) \,,\quad \widehat{\mathfrak{T}}_{\mathbb{Z}_4} = \frac{1}{4}(D_1 + 2D_2 + 2D_3 - D_4 - D_5 + D_6)\,. \tag{161}$$

We have

$$\widehat{\mathfrak{T}}_{\mathbb{Z}_2}^3 = \widehat{\mathfrak{T}}_{\mathbb{Z}_4}^3 = \widehat{\mathfrak{T}}_{\mathbb{Z}_2}^2 \cdot \widehat{\mathfrak{T}}_{\mathbb{Z}_4} = \widehat{\mathfrak{T}}_{\mathbb{Z}_2} \cdot \widehat{\mathfrak{T}}_{\mathbb{Z}_4}^2 = 0 \ (\text{mod } 1)\,. \tag{162}$$

Hence there is no mixed 't Hooft anomaly.

The same result applies to any Spin($2k$) ($k > 4$) on a ($-4$)-curve.

# 6 Topology of the Elliptic Fibrations

Thimbles capture the structure of 1-cycles $H_1(\partial X)$ in the boundary of elliptically fibered Calabi-Yau $n$-folds $X \to B$ which trivialize when included into the bulk. In M-theory M2 branes wrapped on thimbles determine the spectrum of line defects and their Pontryagin dual divisors generate 1-form symmetries. Higher dimensional defects and higher-form symmetries are captured analogously by homology groups $H_k(\partial X)$ which trivialize included into the bulk, sweeping out non-compact cycles in one dimension higher in the process. In this section we study the homology groups $H_k(\partial X)$.

## 6.1 Local K3s

We begin by discussing local K3s which model the elliptic fibration normal to generic points on the discriminant loci of more general $n$-folds. Parts of this discussion appeared in [17].

Table 1: Kodaira's classification of singular fibers, monodromies and torsion subgroups.

| Fiber | Monodromy $T$ | Tor $H_1(\partial X, \mathbb{Z})$ | ADE | Fiber | Monodromy $T$ | Tor $H_1(\partial X, \mathbb{Z})$ | ADE |
|---|---|---|---|---|---|---|---|
| $I_n$ | $\begin{pmatrix} 1 & n \\ 0 & 1 \end{pmatrix}$ | $\mathbb{Z}_n$ | $A_{n-1}$ | $I_n^*$ | $\begin{pmatrix} -1 & -n \\ 0 & -1 \end{pmatrix}$ | $\mathbb{Z}_2 \times \mathbb{Z}_2$ ($n$ even) $\mathbb{Z}_4$ ($n$ odd) | $D_{4+n}$ |
| $II$ | $\begin{pmatrix} 1 & 1 \\ -1 & 0 \end{pmatrix}$ | $0$ | / | $IV^*$ | $\begin{pmatrix} -1 & -1 \\ 1 & 0 \end{pmatrix}$ | $\mathbb{Z}_3$ | $E_6$ |
| $III$ | $\begin{pmatrix} 0 & 1 \\ -1 & 0 \end{pmatrix}$ | $\mathbb{Z}_2$ | $A_1$ | $III^*$ | $\begin{pmatrix} 0 & -1 \\ 1 & 0 \end{pmatrix}$ | $\mathbb{Z}_2$ | $E_7$ |
| $IV$ | $\begin{pmatrix} 0 & 1 \\ -1 & -1 \end{pmatrix}$ | $\mathbb{Z}_3$ | $A_2$ | $II^*$ | $\begin{pmatrix} 0 & -1 \\ 1 & 1 \end{pmatrix}$ | $0$ | $E_8$ |

Consider local K3s $X_2 \to B = \mathbb{C}$ whose discriminant locus consists of single point located at the origin of $B$, which is the setting of section 3.1 where Kodaira thimbles were introduced. The boundary $\partial X$ is smooth and inherits an elliptic fibration $\partial X \to \partial B = S^1$ from the bulk. As all smooth manifolds fibered over a circle its homology groups are therefore determined by the monodromy mappings

$$T_k : \quad H_k(\mathbb{E}) \to H_k(\mathbb{E}), \tag{163}$$

which enter into the short exact sequence

$$0 \to \operatorname{coker}(T_k - 1) \to H_k(\partial X) \to \ker(T_{k-1} - 1) \to 0, \tag{164}$$

derived from the Mayer-Vietoris long exact sequence. Of these mappings $T_1 \equiv T$ differs from the identity. We collect matrix representations of $T$ and the torsion subgroups Tor $H_1(\partial X)$ computed from (164) in table 1. The homology groups of the boundary $\partial X$ compute to

$$H_*(\partial X) = \begin{cases} \{\mathbb{Z}, \mathbb{Z}^2 \oplus \operatorname{Tor} H_1(\partial X), \mathbb{Z}^2, \mathbb{Z}\}, & I = I_n, \\ \{\mathbb{Z}, \mathbb{Z} \oplus \operatorname{Tor} H_1(\partial X), \mathbb{Z}, \mathbb{Z}\}, & I \neq I_n. \end{cases} \tag{165}$$

The case $I = I_n$ is distinguished by the existence of a monodromy invariant 1-cycle. Here $I$ is short for the fiber types collected in table 1. The cokernel in degree one of (164) increases by one in rank compared to other cases.

Finally we describe the two-chains bounding finite copies of the generators of Tor $H_1(\partial X)$. Denote the basis of $H_1(\mathbb{E})$ for which the monodromy matrices take the form given in table 1 by $(A, B)$. Then the bounding two-chains are constructed by fibering the generators of coker$(T - 1)$ over the base circle. For example, for an $I_n$ singularity fibering the $B$-cycle over $S^1$ constructs a two-chain with boundary $nA$. This construction generalizes to $n$-folds as we discuss next.

## 6.2 $I_n$ Singularities

We now generalize the computation of Tor $H_*(\partial X)$ to elliptic $(m+1)$-folds with Weierstrass model $W \to B = \mathbb{C}^m$ and mutually local $I_{n_k}$ singularities. This set-up serves as a toy model for more complicated bases and fibers. Here $k$ labels the irreducible non-compact components of

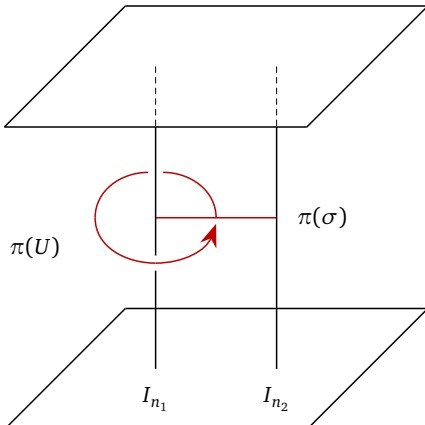

Figure 10: Neighborhood $\pi(U)$ of the cycle $\pi(\sigma)$ with $\sigma$ generating $\mathrm{Tor}\,H_2(\partial X)$. The interval $\pi(\sigma)$ ends on the discriminant loci $I_{n_1}, I_{n_2}$. Let $\gamma$ be the $A$-cycle fibering $\sigma$. Construct a three-chain by fibering the $B$-cycle over a family of loops, each linking $I_{n_1}$ and starting at a distinct point on $\pi(\sigma)$. Along any of these loops the monodromy $B \to B + n_1 A$ acts and therefore the boundary of this three-chain computes to $n_1\sigma$. Similarly construct a three-chain with boundary $n_2\sigma$. It follows that $\gcd(n_1, n_2)\sigma = 0$ in homology.

a connected discriminant $\Delta$. Let $\pi : X \to B$ be the resolved model and denote the vanishing cycle of the geometry by $\gamma$. The base boundary is a sphere and compact cycles $\sigma \in H_k(\partial X)$ with $1 \leq k \leq 2m$ must therefore involve the discriminant in their construction and project such that either

$$\pi(\sigma) \in H_*(\Delta \cap \partial B) \quad \text{or} \quad \partial\,\pi(\sigma) \in H_*(\partial B, \Delta \cap \partial B). \tag{166}$$

In other words, either the base projections or boundaries thereof are contained in the discriminant locus. Let us now specialize to three-folds $m = 2$. In this case the discriminant is further characterized by a split monodromy cover [82] and consequently cycles projecting to the discriminant locus are not acted on by monodromies along paths in $\Delta$ and therefore they are non-torsional. We can further characterize cycles in $H_{k+1}(\partial X)$ with projections bounded by the discriminant as constructed from a cycle in $H_k(\partial B, \Delta \cap \partial B)$ by fibering $\gamma$ over it. Generators of $\mathrm{Tor}\,H_k(\partial X)$ are therefore necessarily fibered by $\gamma$ and associated with relative cycles in the base. We give an example.

**Example: Transverse $I_{n_1}, I_{n_2}$ Intersection.** Consider the three-fold example of an intersecting $I_{n_1}$ and $I_{n_2}$ loci in $B = \mathbb{C}^2 = \mathbb{C}_1 \times \mathbb{C}_2$ tuned on $\mathbb{C}_1 \times \{0\}$ and $\{0\} \times \mathbb{C}_2$ respectively. The two discriminant components intersect transversely at the origin and intersect the boundary three-sphere in a Hopf link $\partial B \cap \Delta = S_{n_1}^1 \cup S_{n_2}^1$. From the above discussion we conclude

$$\mathrm{Tor}\,H_2(\partial X_3) \cong \mathbb{Z}_{\gcd(n_1, n_2)}, \tag{167}$$

with $\mathrm{Tor}\,H_k(\partial X_3) \cong 0$ in all other cases. The generator of $\mathrm{Tor}\,H_2(\partial X_3)$ is the 2-cycle constructed by fibering $\gamma$ over a line with one end on $S_{n_i}^1$ each. We argue for the order of the torsion group by describing the three-chains in $\partial X_3$ bounding $\gcd(n_1, n_2)$ copies of the generator of (167). Denote the generator $\mathrm{Tor}\,H_2(\partial X_3)$ by $\sigma$ and consider a small neighborhood $U$ of $\sigma$. The discriminant restricts to two line segments in $\pi(U)$ and three-chains are constructed following figure 10. The vanishing of torsion degree three follows from Poincaré duality and the universal coefficient theorem which imply $\mathrm{Tor}\,H_1(\partial X_3) \cong \mathrm{Tor}\,H_3(\partial X_3)$ for five-manifolds.

Alternatively, we can compute the homology groups of the boundary $\partial X_3$ using the Mayer-Vietoris sequence. We decompose $\partial X_3$ into a neighbourhood $T$ of the singular fibers and the

complement thereof. We find the result

$$H_*(X_3) \cong \left\{ \mathbb{Z}, \mathbb{Z}, \mathbb{Z}^{n_1+n_2} \oplus \mathbb{Z}_{\gcd(n_1,n_2)}, \mathbb{Z}^{n_1+n_2}, \mathbb{Z}, \mathbb{Z} \right\}. \tag{168}$$

The generators in degree one and four are the $B$-cycle and $\gamma$ fibered over $\partial B = S^3$ respectively. The free parts in degree two and three are introduced by the resolution. The cycle described in figure 10 generates the torsion subgroup in degree two.

The example generalizes straight forwardly to three-folds with $N$-component discriminant loci. Here each irreducible $I_{n_k}$ locus intersects the boundary in a knot $K_{n_k}$. Overall the discriminant intersects the boundary in a link $L = \cup_{k=1}^{N} K_{n_k}$. We have

$$H_1(\partial B, \Delta \cap \partial B) \cong \mathbb{Z}^{N-1}, \tag{169}$$

and the monodromy actions imply $\gcd(n_1, n_2, \ldots, n_N)\gamma = 0$ in $H_1(\partial X_3^\circ)$. Fibering $\gamma$ over the generators of $H_1(\partial B, \Delta \cap \partial B)$ we find

$$\mathrm{Tor}\, H_2(\partial X_3) \cong \mathbb{Z}_{\gcd(n_1,n_2,\ldots,n_N)}^{N-1}, \tag{170}$$

with $\mathrm{Tor}\, H_k(\partial X_3) \cong 0$ in all other cases.

## 6.3 Boundary Topology of Conformal Matter and Single Node Geometries

The examples discussed in the previous section generalize straight forwardly to the geometries engineering conformal matter theories. These are given by transversely intersecting singularities in a $\mathbb{C}^2$ base with tensor branch geometries of the type [83]

$$\left[\mathfrak{g}_L\right] \;—\; C \;—\; \left[\mathfrak{g}_R\right], \tag{171}$$

where $C$ denotes a collection of curves supporting gauge algebras. We immediately conclude that whenever either $\mathfrak{g}_L$ or $\mathfrak{g}_R$ are not engineered by $I_n$ or $I_n^{\mathrm{ns}}$ singularities, then we have

$$\mathrm{Tor}\, H_1(\partial X) = \mathrm{Tor}\, H_3(\partial X) = 0, \tag{172}$$

which simply follows from the resolved fibers having no 1-cycles unless the singularity type is $I_n^s, I_n^{\mathrm{ns}}$. Vanishing of torsion in degree three follows from Poincaré duality and the universal coefficient theorem. Any candidate 1-cycle in the boundary can be deformed to the discriminant locus and collapsed. These vanishing cycles can however sweep out 2-cycles and for $(D_n, D_n)$ or $(E_k, E_k)$ conformal matter we have for example

$$\mathrm{Tor}\, H_2(\partial X_3) \cong \Gamma, \tag{173}$$

where $\Gamma$ is the center of $D_n$ or $E_k$ respectively. This follows from the considerations identical to that for the collision of $I_n$ and $I_m$ components. Wrapping M5 branes on the non-compact 3-cycles intersecting the boundary in the cycles (173), we obtain 3d defect operators in 5d, which are charged under a 3-form symmetry. This is consistent with the global form of the flavor symmetry being the center-quotiented group [81,84], and that gauging a 0-form symmetry in 5d results in a 3-form symmetry.

As our final example consider the geometry

$$\overset{\mathfrak{so}(8+2n)}{4} - [\mathfrak{sp}(2n)]. \tag{174}$$

The base is the total space $B = \mathcal{O}_{\mathbb{P}^1}(-4)$ with Lens space boundary $\partial B = S^3/\mathbb{Z}_4$. The non-compact $I_{2n}^{\mathrm{ns}}$ locus lies along a fiber of $B$ and therefore intersects $S^3/\mathbb{Z}_4$ in a Hopf circle $S_H^1$.

The singular fibers are a necklace of $2n$ two-spheres which contains a 1-cycle $\beta$ running once around the necklace. Traversing the Hopf circle this 1-cycle is mapped to its negative by the fact that the fibration is non-split and we conclude that it contributes a factor of $\mathbb{Z}_2$ to $\mathrm{Tor}\,H_1(\partial X_3)$. We have $H_*(\partial B, \Delta \cap \partial B) = 0$ and therefore $\mathrm{Tor}\,H_2(\partial X_3) = 0$. There exists a two-chain in the base bounding 4 copies of $S_H^1$ and fibering $\gamma$ over this two-chain we construct the 3-cycle $\alpha$ generating $\mathrm{Tor}\,H_3(\partial X_3) = \mathbb{Z}_2$. Here $\gamma$ is the 1-cycle collapsing in the singular fibers. The fact that $2\alpha = 0$ follows from the $U(1)$ action on the Lens space generated by flow along the Hopf fibers. Let us decompose $S^3/\mathbb{Z}_4$ into a disjoint union of Hopf circles. Each point on $\alpha$ lies on a Hopf fiber and is moved once along the Hopf fibers returning to its original position. These Hopf circles both link the non-compact $I_{4n}^{\mathrm{ns}}$ and the compact $I_n^*$ loci which results in the total monodromy acting as $\gamma \to -\gamma$ and therefore $\alpha \to -\alpha$, consequently $2\alpha = 0$. The Hopf fiber in the base contributes a factor of $\mathbb{Z}_4$ to $\mathrm{Tor}\,H_1(\partial X_3)$ and the 3-cycle constructed by fibering the singular fibers over the Hopf cycle contributes $\mathbb{Z}_4$ to $\mathrm{Tor}\,H_3(\partial X_3)$. Overall we find

$$\mathrm{Tor}\,H_1(\partial X_3) \cong \mathbb{Z}_2 \oplus \mathbb{Z}_4\,, \qquad \mathrm{Tor}\,H_2(\partial X_3) \cong 0\,, \qquad \mathrm{Tor}\,H_3(\partial X_3) \cong \mathbb{Z}_2 \oplus \mathbb{Z}_4\,, \qquad (175)$$

which is consistent with our considerations based on thimbles. Again the absence of the 3-form symmetry is consistent with the global flavor symmetry group.

# 7 Discussion and Outlook

In this paper we studied the defect group of supersymmetric quantum field theories engineered by elliptically fibered Calabi-Yau manifolds in M/F-theory. Our main focus was on theories which admit polarizations to an absolute theory with a 1-form symmetry. Genuine lines in such theories are characterized by non-compact relative 2-cycles of the $n$-fold. We argued that such 2-cycles are grouped into equivalence classes, referred to as Kodaira thimbles, by screening effects in codimension 1, that is by screening with curves ruling Cartan divisors. Additional screening effects enter through compact curves/local operators supported at the codimension 2 locus and these give rise to dependency relations among Kodaira thimbles which determine the defect group of lines and the Pontryagin dual 1-form symmetry.

Crucial in quantifying these effects was the identification of Kodaira thimbles with a rational collection of compact 2-cycles with coefficients mod 1. Such representations immediately permitted us to introduce divisors Pontryagin dual to Kodaira thimbles. These divisor in turn are key to geometrizing the 1-from symmetry generators. Their intersection numbers determine the topological couplings for an associated SymTFT capturing the (mixed) 't Hooft anomalies of higher-form symmetries.

We illustrated these general insights in a large class of examples. Elliptic fibrations with codimension 2 singular fibers, i.e. matter, result in additional screening effects, which were analyzed for both transversely and non-transversely intersecting discriminant loci. The SymTFT couplings were computed for all single node NHCs and Spin$-Sp$ quivers.

Finally we considered the torsion subgroups of the boundary related to higher than 1-form symmetry groups. For elliptic three-folds there are two independent subgroups, the first characterizing 1-form symmetries and their dual 2-form symmetries and the second determining possible 0-form symmetries and dual 3-form symmetries. We computed these groups for Spin$-Sp$ and conformal matter theories.

The framework that was developed in this paper can be applied to elliptic fibrations in any complex dimension. We discussed elliptically fibered two- and three-folds, but a similar analysis will be applicable in four-folds and five-folds, realizing 4d $\mathcal{N} = 1$ and 2d $(0, 2)$ theories, respectively. As we stated in the main text, we do not expect further screening of the line operators in four-folds, as the higher codimension singular fibers will correspond to su-

perpotential couplings. For five-folds, which correspond to compactifications to 2d [85–88], we expect some interesting effects due to the small dimensionality of the spacetime. It would be interesting to construct the full category of lines in these 2d (0, 2) theories.

Another obvious application of this framework is the study of 2-groups in 5d and 6d [11, 12, 21]. We expect the combination of results in [62, 63], where the non-trivial extension group, that underlies the 2-group symmetry, is identified in terms of boundary topological data. Combined with the results of this present work, this should lead to a comprehensive understanding of 2-groups in F-theory compactifications, and should match the intersection theoretic approach underlying the classification of 2-group symmetries in 6d SCFTs [11]. In particular the thimbles ending on the singular boundary of the compactification space, will have non-trivial relations that map them into flavor Wilson lines. We will report on this in future work.

# Acknowledgements

We thank F. Apruzzi, L. Bhardwaj, F. Bonetti, M. del Zotto, I. García Etxebarria, for discussions on related matters. MEH, DRM, SSN acknowledge support from the Simons Collaboration grant "Special Holonomy in Geometry, Analysis, and Physics" under grant numbers 724069, 488629, and 724073, respectively. SSN is supported in part by the European Union's Horizon 2020 Framework: ERC grant 682608. YNW is supported by National Science Foundation of China under Grant No. 12175004 and by Peking University under startup Grant No. 7100603534.

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
