# Peer review of "Generalized Symmetries in F-theory and the Topology of Elliptic Fibrations"

_SciPost Physics, doi:SciPost Phys. 13, 030 (2022)_

## Round 2 · Referee Report · Anonymous (Referee 1) · 2022-5-20

Strengths

The authors developed some new and interesting F-theory techniques that can be applied to the study of higher-form symmetries of 6 dimensional field theories.

Weaknesses

The style of writing in its current form is not very physical, not very accessible, and not very "readable" to quantum field theorists who are not an experts on F-theory. The authors could have made more effort to improve the "readability" of the article.

Report

The authors of this article developed certain techniques in F-theory to study higher form symmetries and their mixed anomalies in some 6d and the corresponding 5d Kaluza-Klein theories. In particular, the correspondence between the Kodaira thimbles and the group of line operators (defect group) are pointed out in this article. The authors then applied this correspondence to compute the defect group of several non-Higgsable clusters for 6 dimensional SCFTs. The mixed anomalies of higher-form symmetries were then determined from the triple intersection numbers. Since this article has some interesting material, I recommend this article for publication in the SciPost after addressing the comments on the "Requested Changes" section.

Requested changes

  1. In the introduction of the article, I would recommend the authors to spell out further the physical meaning of the following phrase: "the discriminant of the elliptic fibration intersects the boundary". It would be useful if the authors could give some explicit physical examples of theories that have this property and those that do not have this property. This would serve as a nice motivation for the study of the material in this article.
  2. The 6 dimensional conformal matter theories discussed in this article contain one gauge group. Can this be generalised to those with many gauge groups?
  3. Can the technique developed in this article be applied to detect the mixed anomaly that arises from gauging the 1-form symmetry that participates in the 2-group structure, for example, those discussed in Section 3.4 of [arXiv:2110.14647]? If so, could the authors explain how? And if not, could the authors explain what the problem is?

  • validity: high
  • significance: good
  • originality: high
  • clarity: good
  • formatting: -
  • grammar: -

Author:  Max Hubner  on 2022-07-20  [id 2671]

(in reply to Report 1 on 2022-05-20)
Category:
remark
answer to question
correction

We thank both referees for their valuable feedback and suggestions.

Regarding the first point raised in the this report, we have now made modifications in the introduction highlighting that the non-compact discriminant components intersecting the boundary have the interpretation of flavor branes and that their effect on the boundary topology geometrizes the screening effects of matter fields.

The second remark in the report concerns the question whether our methods apply in the context of semi-simple gauge groups. Here we point out that the general discussion makes no assumption on whether gauge groups are simple or semi-simple. Indeed, the examples discussed in section 4 of the paper which cover cases such as SU(n)xSU(m) gauge theories, also demonstrate our formalism in such cases.

Regarding the final comment on studying further anomalies involving flavor symmetries we point out that such questions lie outside the scope of the presented paper. Such computations would require geometrizing the global form of the flavor symmetry (as discussed in later papers such as 2203.10097 and 2203.10102) and involve methods not developed in this paper -- purposefully as these two papers were already in progress, and build on the current paper.

We hope that the revisions have now made the paper suitable for publication.

---

## Round 2 · Referee Report · Anonymous (Referee 2) · 2022-5-29

Strengths

1 Mathematically clear exposition and general statements of new geometrical methods to work out higher form symmetries in F-Theory compactifications

2 Detailed discussion of a number of examples

Weaknesses

1 No real introduction to any of the concepts used, so only understandable by people already well versed in the field. Some aspects are hard to follow and further details might be useful. E.g. statement of the long exact sequence and definition of maps used in 2.1 or a reference; definition of \mathbb{E};

Report

The literature on higher-form symmetries (HFS) from M-theory so far only covered QFTs engineered using non-compact geometries with localized singularities, i.e. those not intersecting the boundary. When engineering QFTs in F-theory this is not the typical situation (e.g. in cases with flavour symmetry) and the above methods need to be extended to singular geometries. This generalization is what this works aims to address. This is particularly relevant as the topology of the boundary is crucial in determining the HFS. Equally relevant, this work explains how to work out HFS by using the natural data in terms of which F-Theory compactifications are typically presented. The crucial tool here are Lefschetz thimbles, i.e. one-cycles of the elliptic fibration which are collapsed somewhere on the discriminant locus. The authors give a range of instrutive examples and even discuss how to work out couplings in the associated symmetry TFT.

This work is a great exploration of how to work out HFS from the topology of Lefschetz thimbles. This new approach opens the door for in-depth studies of a great variety of QFTs that can be engineered via F-Theory.

Requested changes

none

---

## Round 3 · Referee Report · Anonymous (Referee 1) · 2022-7-20

Report

The authors made some minor improvements on the draft. The "readability" of the paper is still low. Nevertheless it could be published in this current form.

---

## Round 3 · List of Changes

addressed referee comments

---

## Editorial Decision

published